nature
ecology & evolution
# Ancient genomes from the last three millennia support multiple human dispersals into Wallacea

Sandra Oliveira [1,24] ✉, Kathrin Nägele[2,24], Selina Carlhoff [2,24], Irina Pugach[1], Toetik Koesbardiati[3], Alexander Hübner [1,2], Matthias Meyer [1], Adhi Agus Oktaviana[4], Masami Takenaka [5], Chiaki Katagiri[6], Delta Bayu Murti[3], Rizky Sugianto Putri[3], Mahirta[7], Fiona Petchey [8,9], Thomas Higham[10,11], Charles F. W. Higham[12], Sue O'Connor [13,14], Stuart Hawkins [13,14], Rebecca Kinaston[15,16,17], Peter Bellwood[18], Rintaro Ono[19], Adam Powell [20], Johannes Krause [2,24,25] ✉, Cosimo Posth [2,21,22,24,25] ✉ and Mark Stoneking [1,23,24,25] ✉

Previous research indicates that human genetic diversity in Wallacea—islands in present-day Eastern Indonesia and Timor-Leste that were never part of the Sunda or Sahul continental shelves—has been shaped by complex interactions between migrating Austronesian farmers and indigenous hunter–gatherer communities. Yet, inferences based on present-day groups proved insufficient to disentangle this region's demographic movements and admixture timings. Here, we investigate the spatio-temporal patterns of variation in Wallacea based on genome-wide data from 16 ancient individuals (2600–250 years BP) from the North Moluccas, Sulawesi and East Nusa Tenggara. While ancestry in the northern islands primarily reflects contact between Austronesian- and Papuan-related groups, ancestry in the southern islands reveals additional contributions from Mainland Southeast Asia that seem to predate the arrival of Austronesians. Admixture time estimates further support multiple and/or continuous admixture involving Papuan- and Asian-related groups throughout Wallacea. Our results clarify previously debated times of admixture and suggest that the Neolithic dispersals into Island Southeast Asia are associated with the spread of multiple genetic ancestries.

Wallacea (Fig. 1), a region of deep-sea islands located between the Sunda and Sahul continental shelves[1], has been both a bridge and a barrier for humans migrating from Asia to Oceania. Anatomically modern humans (AMHs) presumably first crossed Wallacea before reaching Sahul, for which the earliest unequivocal dates are approximately 47 ka[2–5] (but see Clarkson et al.[6]). In Wallacea itself, the archaeological record indicates occupation by AMHs starting around 46 ka in the southern islands[7–9], 45.5 ka in Sulawesi[10] and 36 ka in the northern islands (North Moluccas)[11]. After a long period of isolation, the region was impacted by the Austronesian expansion. Equipped with new sailing and farming technologies, Austronesian-speaking groups likely expanded out of Taiwan 4,000–4,500 ya[12–14] and eventually settled in Island Southeast Asia (ISEA), Oceania and Madagascar. Their arrival is generally linked to the earliest appearance of pottery, which dates to approximately 3,500 ya in Wallacea[11,15–18]. During the Late Neolithic and early Metal Age (2,300–2,000 ya), the maritime trade network intensified, with a movement of spices, bronze drums and glass beads connecting Wallacea to India and mainland SEA (MSEA)[11,17,19–24].

The contact between Austronesian-speaking farmers and hunter–gatherer communities is still reflected in the linguistic and biological diversity of Wallacea today. Austronesian languages of the Malayo-Polynesian subgroup are widespread throughout the

[1]Department of Evolutionary Genetics, Max Planck Institute for Evolutionary Anthropology, Leipzig, Germany. [2]Department of Archaeogenetics, Max Planck Institute for Evolutionary Anthropology, Leipzig, Germany. [3]Department of Anthropology, Faculty of Social Sciences and Political Sciences, Universitay Airlangga, Surabaya, Indonesia. [4]The National Research Center for Archaeology, Jakarta, Indonesia. [5]Kagoshima Women's College, Kagoshima, Japan. [6]Okinawa Prefectural Archaeological Center, Nishihara, Japan. [7]Jurusan Arkeologi, Fakultas Ilmu Budaya, Universitas Gadjah Mada, Yogyakarta, Indonesia. [8]Radiocarbon Dating Laboratory, University of Waikato, Hamilton, New Zealand. [9]ARC Centre of Excellence for Australian Biodiversity and Heritage, College of Arts, Society and Education, James Cook University, Cairns, Queensland, Australia. [10]Department of Evolutionary Anthropology, University of Vienna, Vienna, Austria. [11]Oxford Radiocarbon Accelerator Unit, Research Laboratory for Archaeology and the History of Art, University of Oxford, Oxford, UK. [12]Department of Anthropology, University of Otago, Dunedin, New Zealand. [13]School of Culture, History and Language, College of Asia and the Pacific, Australian National University, Acton, Australian Capital Territory, Australia. [14]Australian Research Council Centre of Excellence for Australian Biodiversity and Heritage, Australian National University, Canberra, Australian Capital Territory, Australia. [15]Department of Anatomy, School of Medical Sciences, University of Otago, Dunedin, New Zealand. [16]Griffith Centre for Social and Cultural Research, Griffith University, Southport, Queensland, Australia. [17]BioArch South, Waitati, New Zealand. [18]School of Archaeology and Anthropology, College of Arts and Social Sciences, Australian National University, Canberra, Australian Capital Territory, Australia. [19]Center for Cultural Resource Studies, National Museum of Ethnology, Osaka, Japan. [20]Department of Human Behavior, Ecology and Culture, Max Planck Institute for Evolutionary Anthropology, Leipzig, Germany. [21]Institute for Archaeological Sciences, Archaeo- and Palaeogenetics, University of Tübingen, Tübingen, Germany. [22]Senckenberg Centre for Human Evolution and Palaeoenvironment, University of Tübingen, Tübingen, Germany. [23]Université Lyon 1, Centre National de la Recherche Scientifique, Laboratoire de Biométrie et Biologie Evolutive, Villeurbanne, France. [24]These authors contributed equally: Sandra Oliveira, Kathrin Nägele, Selina Carlhoff. [25]These authors jointly supervised this work: Johannes Krause, Cosimo Posth, Mark Stoneking. ✉e-mail: sandra_oliveira@eva.mpg.de; krause@eva.mpg.de; cosimo.posth@uni-tuebingen.de; stonekg@eva.mpg.de

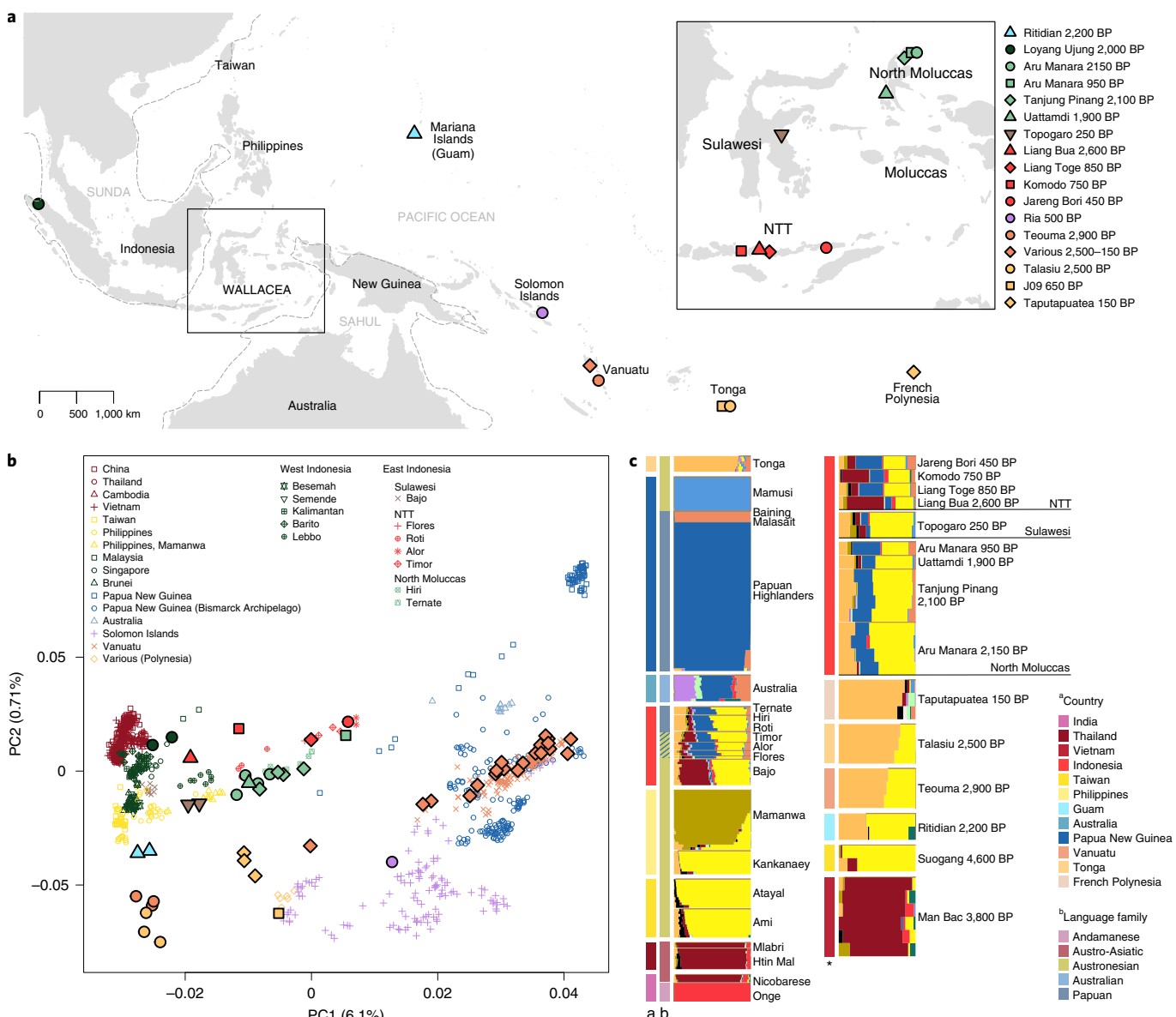

**Fig. 1 | Sample provenance and the results of principal component and DyStruct analyses. a**, Map showing the location of ancient individuals. **b**, PCA of publicly available whole-genome data merged with the Affymetrix Human Origins and Affymetrix 6.0 genotype data (dataset 1). Ancient individuals (shown with a black contour) are projected and their fill colour matches the colour of present-day individuals from the same geographical area. **c**, DyStruct results for dataset 1 displaying only a subset of the individuals included in the full analysis (Supplementary Fig. 1e). Newly generated individuals are highlighted in bold in the legend. Country and language information are displayed as colour bars to the left of the inferred ancestry components.

region[25] but a few dozen non-Austronesian (that is, Papuan) languages are also spoken in the North Moluccas, Timor, Alor and Pantar[26]; some Austronesian languages show features acquired from Papuan languages[27].

The genomic composition of present-day Wallaceans shows signals of admixture between Papuan- and Asian-related ancestry most similar to that of present-day Austronesians[28–30]. This dual ancestry is geographically distributed as a gradient of increasing Papuan-related ancestry from west to east[28,30]. Previous studies have estimated admixture times based on present-day groups[28–30], providing the first inferences on the direction and rate of spread of genetic ancestry[28]. However, the time estimates from different studies show discrepancies of more than 3,000 years (Supplementary Table 1) that cannot be solely attributed to ascertainment bias but also reflect limitations in admixture dating methods[28,29], which are

differentially affected by scenarios involving continuous or multiple pulses of gene flow from closely related sources[31]. Resolving the uncertainty in admixture dates has important implications for understanding the interactions between Austronesians and pre-Austronesian populations. Admixture dates close to the archaeological dates proposed for the Austronesian arrival would indicate that admixture occurred soon after contact, while more recent dates would imply that communities coexisted for some time before genetically mixing or were mixing for a prolonged period. Moreover, admixture dates predating the Austronesian arrival would suggest alternative explanations, such as genetic influences from other Asian-related groups[32].

In this study, we leveraged the power of ancient DNA to investigate spatio-temporal patterns of variation within Wallacea during the last 2,500 years. We provide insights into the time of arrival of

**1025**

**Table 1 | Ancient samples from Wallacea included in this study**

| Sample name | Island, region | ¹⁴C date ± s.d. (BP) | Assigned group | Sex | mtDNA haplogroup | Y-chromosome haplogroup | Number of SNPs |
|---|---|---|---|---|---|---|---|
| AMA001 | Morotai, North Moluccas | 2,258 ± 30 | Aru Manara 2,150 BP | M | B4a1a1 | *C1b1a2b* | 255,458 |
| AMA003008 | Morotai, North Moluccas | 2,130 ± 24 | Aru Manara 2,150 BP | F | *Q1d* | - | 576,009 |
| AMA004 | Morotai, North Moluccas | 2,009 ± 24 | Aru Manara 2,150 BP | F | M73a | - | 933,715 |
| AMA005 | Morotai, North Moluccas | n/a | Aru Manara 2,150 BP | F | B4a1a1 | - | 209,933 |
| AMA009 | Morotai, North Moluccas | 968 ± 20 | Aru Manara 950 BP | F | *Q1d* | - | 935,157 |
| TanjungPinang1 | Morotai, North Moluccas | 2,090 ± 180 | Tanjung Pinang 2,100 BP | M | *Q* | *S1d1~* | 870,652 |
| TanjungPinang2 | Morotai, North Moluccas | n/a | Tanjung Pinang 2,100 BP | M | *Q1* | O2a2b2a2b2 | 939,665 |
| TanjungPinang4 | Morotai, North Moluccas | n/a | Tanjung Pinang 2,100 BP | M | *Q* | *S1a1b1d2b~* | 897,982 |
| TanjungPinang6 | Morotai, North Moluccas | n/a | Tanjung Pinang 2,100 BP | M | B4a1a | O2a2b2a2b2 | 1,028,190 |
| Uattamdi1 | Kayoa, North Moluccas | 1,915 ± 27 | Uattamdi 1,900 BP | M | E1a1a1 | O1a2a1 | 895,300 |
| TOP002 | Sulawesi, central Sulawesi | 211 ± 24 | Topogaro 250 BP | M | E2a | O2a2a1a2a2 | 697,028 |
| TOP004 | Sulawesi, central Sulawesi | 324 ± 24 | Topogaro 250 BP | M | E2a | *M1a* | 249,209 |
| KMO001 | Komodo, NTT | 726 ± 19 | Komodo 750 BP | F | B4a1a1 | - | 122,610 |
| LIA001002 | Flores, NTT | 2,588 ± 23 | Liang Bua 2,600 BP | F | M17a | - | 873,614 |
| LIT001 | Flores, NTT | 861 ± 20 | Liang Toge 850 BP | F | E1a2 | - | 623,960 |
| JAB001 | Pantar, NTT | 457 ± 19 | Jareng Bori 450 BP | F | M7b1a2a1 | - | 848,849 |

The reported radiocarbon dates (¹⁴C) are uncalibrated. MtDNA and Y-chromosome haplogroups whose origins have been previously associated with Papuan groups are in italics; all other haplogroups are most likely of Asian-related origin.

the Austronesian-related ancestry, the temporal span of admixture and the relationship between the ancestry of incomers and that of other groups from Asia and Oceania. Additionally, we explore the impact and timing of an additional migration from MSEA to Wallacea.

## Results

We extracted DNA from skeletal remains from 16 individuals dated to approximately 2,600–250 BP from 8 archaeological sites spanning the North Moluccas, Sulawesi and East Nusa Tenggara (for our purposes, East Nusa Tenggara has been abbreviated to NTT for Nusa Tenggara Timur) (Fig. 1a and Table 1). Sequencing libraries were then constructed and capture-enriched for approximately 1.2 million genome-wide single-nucleotide polymorphisms (SNPs)[33] and the complete mitochondrial DNA (mtDNA). The authenticity of ancient DNA was confirmed based on the elevated amounts of deaminated positions at the ends of reads and the short average fragment size (Supplementary Table 2). Contamination estimates were low (Supplementary Table 2).

The mtDNA and Y-chromosome haplogroups show that both Asian- and Papuan-related ancestries were already present in the North Moluccas approximately 2,150 BP (Table 1). Furthermore, 2 North Moluccas individuals dating to approximately 2,150–2,100 BP

carried mtDNA and Y-chromosome haplogroups associated with different ancestries, indicating that admixture started before then. In comparison to the individuals from NTT and Sulawesi, those from the North Moluccas showed a higher proportion of mtDNA lineages connecting them to Near Oceania, as attested by the Q haplogroups characteristic of Northern Sahul[34] and by the so-called 'Polynesian pre-motif' (B4a1a/B4a1a1) (ref. [35]). None of the individuals from Sulawesi or NTT carry Papuan-related mtDNA haplogroups, even though they are found there today[36,37].

To explore the genome-wide patterns of variation in ancient Wallaceans, we performed principal component analysis (PCA) based on different sets of present-day populations from Asia and Oceania and two combinations of SNP arrays (Methods). Ancient Wallaceans cluster between Papua New Guinea and Asia, together with present-day Wallaceans (Fig. 1b and Extended Data Fig. 1). However, the trajectory outlined by individuals from the northern (North Moluccas) versus southern (NTT) islands is slightly different, suggesting they may have distinct genetic histories. Ancient individuals from NTT cluster on a cline towards mainland Asians and some Western Indonesian groups, while ancient individuals from the North Moluccas align on a trajectory towards present-day Taiwanese/Philippine populations or even towards ancient individuals from Guam 2,200 BP, Vanuatu 2,900 BP and Tonga 2,500 BP

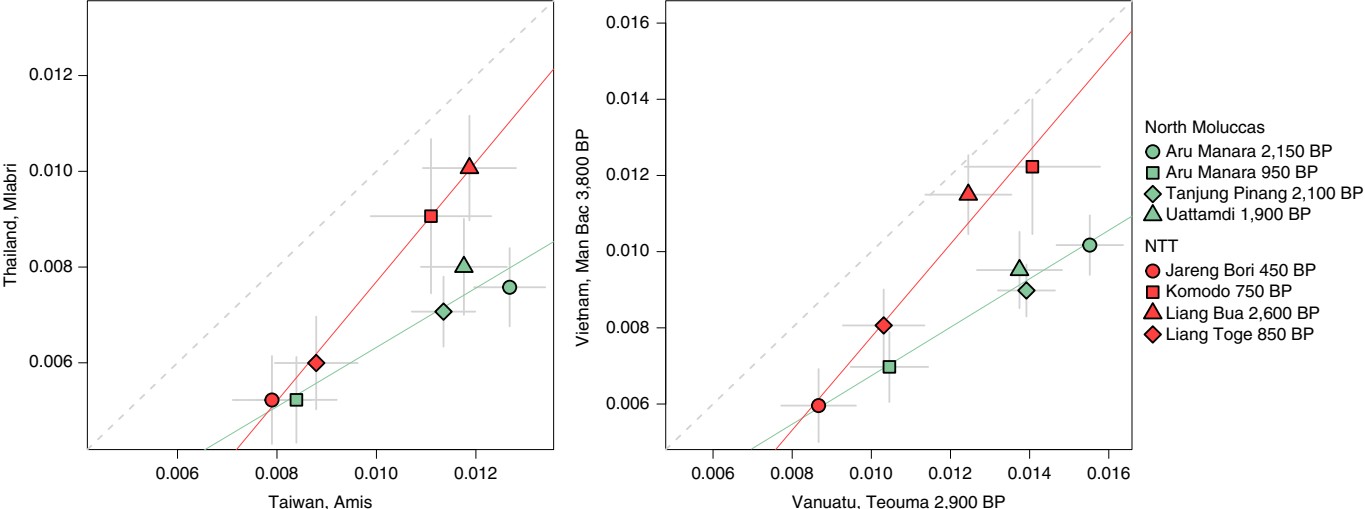

**Fig. 2 | Biplots showing the results of two pairs of $f_4$-statistics of the form $F_4$(Mbuti, test; New Guinea Highlanders, ancient Wallacea).** The test groups are shown on the x and y axis labels. Data are presented as exact $F_4$-values ± 2 s.e. indicated by the grey lines. Linear regression lines for the individuals from the North Moluccas and NTT are shown in green and red, respectively. The results for all tested pairs are shown in Supplementary Figs. 3 and 4; the results of their respective Bayesian models are shown in Supplementary Figs. 5–8.

(previously shown to have almost exclusively Austronesian-related ancestry)[38,39]. The differences between the two Wallacean regions are more pronounced when projecting the ancient individuals into principal components that feature Asian-related variation (PC2 versus PC3; Extended Data Fig. 2).

We next used a model-based clustering method (DyStruct) to infer shared ancestry[40]. The results for the best supported number of clusters in each of the tested datasets (Supplementary Fig. 1a,b) show that ancient Wallaceans shared ancestry with Papuan-speaking groups from New Guinea (dark blue component) and multiple Asian groups whose ancestry can be partitioned into three main components (Fig. 1c; see full results in Supplementary Fig. 1e,f). One component (yellow) is present at high frequencies in Austronesian-speaking groups from Taiwan, the Philippines and Indonesia, and ancient individuals from Taiwan; a second component (mango) is maximized in Polynesian-speaking groups from the Pacific and ancient individuals from the same region; and a third component (dark red) is widespread in present-day and ancient individuals from SEA. The most striking difference among ancient Wallaceans is the presence of the SEA component in ancient NTT and Sulawesi individuals but not in North Moluccan individuals. A more subtle difference occurs in the relative proportion of the two Austronesian-related components (Extended Data Fig. 3): ancient individuals from Sulawesi and NTT have a higher relative proportion of the Austronesian-related (yellow) component that predominates in Taiwan, compared to ancient individuals from the North Moluccas, who are more similar to groups from the Pacific.

To directly compare allele-sharing between ancient Wallaceans and different Asian-related groups, we used $f$-statistics[41]. First, we computed an $f_4$-statistic of the form $F_4$(Mbuti, ancient Wallacean; Amis, test), where the test group includes ancient and present-day groups from mainland Asia, ISEA and the Pacific who have no discernible Papuan-related ancestry (Supplementary Fig. 2 and Supplementary Table 3). Our results show that ancient individuals from the North Moluccas share more drift with ancient individuals from Vanuatu (2,900 BP) and Tonga (2,500 BP) than with Amis ($z > 2$). In contrast, ancient individuals from Sulawesi and NTT do not share additional drift with any tested groups. Nonetheless, the higher number of $f_4$-statistics consistent with zero in tests involving ancient individuals from NTT (Komodo and Liang Bua)

indicates that they share as much drift with Amis as with several other groups, not only from Taiwan/Philippines but also SEA or Western Indonesia. This result, together with the identification of an ancestry component related to SEA (Supplementary Fig. 1e,f) in ancient NTT and Sulawesi individuals, supports a more complex admixture history in these parts of Wallacea.

We next analysed pairs of $f_4$-statistics designed to capture any differences in Asian-related ancestry between individuals from the North Moluccas and NTT (Supplementary Figs. 3 and 4 and Supplementary Tables 4 and 5). All $f_4$-statistics had the form $F_4$(Mbuti, test; New Guinea Highlanders, ancient Wallacean) and each pair compared the results for a fixed test group on the x axis (Amis or Vanuatu 2,900 BP for comparisons between modern or ancient test pairs, respectively) and various Asian-related test groups on the y axis. Since individuals from the North Moluccas lacked the SEA component (Fig. 1c), the best proxy for this component in individuals from NTT should maximize the differences in $f_4$-statistics between regions. To estimate these differences, we used a Bayesian approach that accounts for measurement error in the f₄-statistics (Supplementary Figs. 5 and 6). We concluded that the groups maximizing differences between ancient individuals from Wallacea are the present-day Mlabri or Nicobarese and ancient individuals from Vietnam (Mán Bạc 3,800 BP), Laos (Tam Pa Ping 3,000 BP) and Thailand (Ban Chiang 2,600 BP) (Fig. 2), but other related MSEA proxies could not be excluded (Supplementary Figs. 7 and 8). For several MSEA test groups, the 95% credible interval of the differences did not overlap zero within the range of $F_4$ values covering the Aru Manara 2,150 BP, Tanjung Pinang 2,100 BP, Uattamdi 1,900 BP, Liang Bua 2,600 BP and Komodo 750 BP, indicating strong support for the differences between NTT and the North Moluccas. Below this range (that is, for lower $F_4$ values), there is no support for regional differences, probably due to a decrease in power for differentiating Asian ancestries when the total Asian ancestry is low and the Papuan-related ancestry is high.

We also investigated the relationships between ancient Wallaceans and groups associated with the first colonization of Sahul or Wallacea using an $f$-statistic of the form $F_4$(Mbuti, new ancient Wallacean; New Guinea Highlanders, test). Most ancient Wallacean individuals showed a significantly closer affinity to New Guinea Highlanders than to Australians, the Bismarck group or the

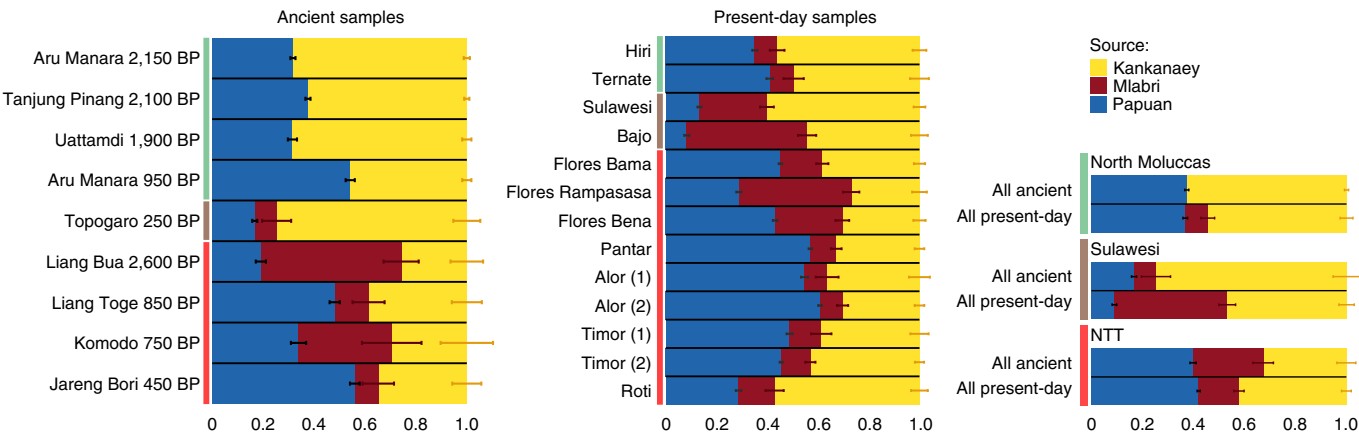

**Fig. 3 | Ancestry proportions estimated with qpAdm for the model with the highest *P* value in each group.** Individuals from the North Moluccas, Sulawesi and NTT are marked as green, brown and red vertical bars, respectively. Horizontal bars show the ancestry proportions ± 1 s.e. (calculated with block jackknife). The number after the name of present-day individuals indicates the genotyping array used: (1) Affymetrix 6.0; (2) Affymetrix Axiom Genome-Wide Human.

recently published pre-Neolithic individual from Sulawesi (Leang Panninge)[42] (Supplementary Fig. 9 and Supplementary Table 6). The non-significant results are probably due to low amounts of data available for Leang Panninge and/or low amount of Papuan ancestry in Liang Bua and Topogaro (Supplementary Fig. 10). Nonetheless, tests involving Leang Panninge consistently exhibited the lowest $F_4$ values. Therefore, despite being from Wallacea, this ancient individual was not a good proxy for the Papuan-related ancestry of the newly reported ancient Wallaceans.

We further investigated potential differences in ancestry sources and proportions among ancient Wallaceans using the qpAdm software[41]. Our results indicate that whereas ancient individuals from the North Moluccas can be modelled as having both Papuan- and Austronesian-related ancestry, ancient individuals from NTT and Sulawesi were either consistent with or required a three-wave model, with additional SEA-related ancestry (Fig. 3 and Supplementary Table 7). Despite cases for which we identified more than 1 fitting model ($P > 0.01$), the estimated proportions under the model with the highest *P* value correlated with the proportions of Austronesian, Papuan and SEA ancestry inferred by DyStruct (Mantel statistic $r = 0.97$, $P < 0.001$). Ancient NTT individuals displayed more inter-island variance in their Papuan- and SEA-related ancestries ($s^2 = 0.026$ and $0.046$, respectively) compared to their Austronesian-related ancestry ($s^2 = 0.003$).

A comparison between the ancestry composition of ancient and present-day individuals from the same region (Fig. 3 and Supplementary Table 7) suggests that a small part (8%) of the Austronesian-related ancestry of ancient individuals from the North Moluccas was replaced by SEA ancestry in present-day groups, masking former differences between regions of Wallacea. The present-day groups from Sulawesi and NTT can be modelled by the same three ancestry components found in ancient individuals from those regions. However, the ancestry proportions of ancient and present-day groups showed some differences, which could indicate ancestry shifts over time or reflect the small sample sizes.

To gain insights into the relative order of admixture events between different ancestries in Wallacea, we used the admixture history graph (AHG) approach[43], which relies on differences in covariance between the components inferred by DyStruct (Supplementary Tables 8–10). The AHG, applied to both ancient and present-day data from NTT, suggests that the admixture of SEA- and Papuan-related ancestries occurred before the arrival of the Austronesian-related ancestry (Supplementary Table 10). An analogous test based on the three main ancestry components

observed in the North Moluccas (Papuan, Taiwan-Austronesian and Pacific-Austronesian) does not provide compelling evidence for backflow from the Pacific since the AHG inferred that Papuan ancestry was introduced into a population that already had both Austronesian-related components (Supplementary Table 10). This result suggests that drift had a more important role in the occurrence and distribution of the two Austronesian-related components.

Finally, we investigated the timing of admixture using the software DATES (Supplementary Table 11), applicable to ancient DNA from single individuals[44]. With time series ancient data, we expected to reconcile previous admixture time estimates, despite gene flow complexity. Using Papuans and a pool of Asian groups as sources, we found that estimates for the oldest individuals from the Northern Moluccas (2,150 BP) and NTT (2,600 BP) are very similar (approximately 3,000 BP, adjusting for the archaeological age of each sample; Fig. 4), approaching archaeological dates for the arrival of the Austronesians in Wallacea. However, younger individuals displayed more recent estimates. This trend of decreasing admixture times extends to the present-day groups from the North Moluccas who show even younger admixture dates (approximately 1,400 BP) than ancient individuals from the same region. Admixture times for present-day and ancient samples from NTT overlapped. Our results indicate that both regions probably experienced multiple admixture pulses (and/or continuous gene flow), as suggested by the changes in ancestry proportions or composition over time (Fig. 3); however, the overall duration of admixture differed between regions.

Since we inferred that the Asian-related ancestry of ancient individuals from NTT was introduced by two Asian groups in separate events, we might expect admixture time estimates to differ using different proxies for this Asian ancestry. Therefore, we looked for differences in estimates using as sources Papuans and Austronesians (test a) versus Papuans and MSEA (test b). While the point estimates for the two NTT samples with the most MSEA ancestry were older in test b than test a, the confidence intervals overlapped (Extended Data Fig. 4). Either the different Asian ancestry sources were too similar or the admixture times were too close in time to reliably distinguish.

## Discussion

This study greatly increases the amount of ancient genomic data from ISEA, a tropical region unsuitable for DNA preservation but particularly important for understanding human population interactions. The new data clarify Wallacea's admixture history and expose genetic relationships that were masked by recent demographic

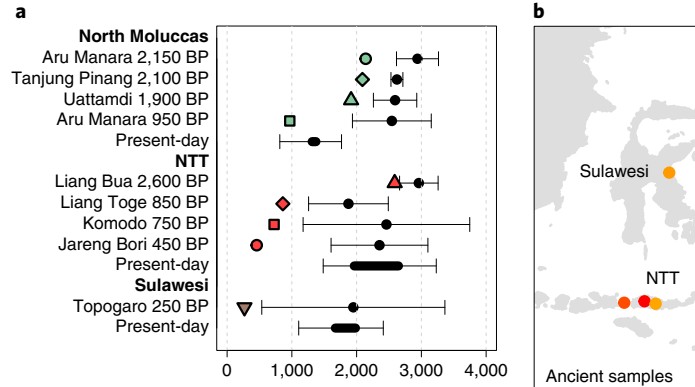
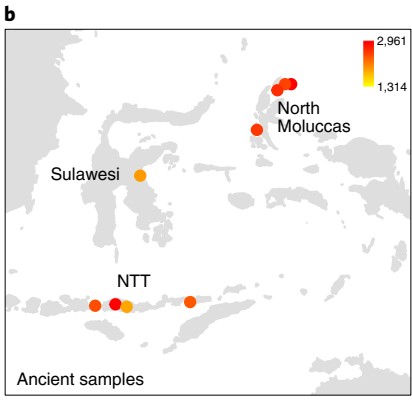
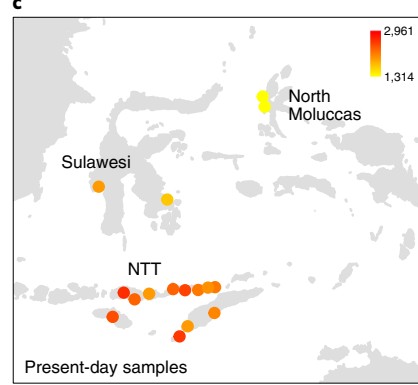

**Fig. 4 | Admixture estimates. a**, Admixture date point estimates ± 2 s.e. are shown in black. For the present-day groups, the bar represents the minimum and maximum point estimates (±2 s.e.). The age of ancient individuals is indicated with filled symbols: green, North Moluccas; brown, Sulawesi; red, NTT. **b,c**, Spatial distribution of admixture date estimates for ancient and present-day individuals, with admixture dates depicted according to the heat plot.

processes in present-day populations. Our results reveal striking regional variation in Wallacea, some of which is found among the Austronesian-related ancestry of ancient individuals. The most remarkable differences are associated with ancestry contributions from MSEA that were probably already part of the NTT genomic landscape when Austronesians arrived but were absent from the North Moluccas until recently.

**Papuan ancestry in Wallacea.** All newly presented ancient Wallaceans are genetically closer to present-day Papuans than to the pre-Neolithic Leang Panninge individual from Sulawesi[42], suggesting little direct continuity between pre-Neolithic and post-Austronesian Wallaceans. Additionally, the ancestry of the newly presented Wallaceans is closer to the ancestry of Papuans than indigenous Australians. This suggests that either the group that gave rise to Australians split first or there was contact between Wallacea and New Guinea after their initial settlement. The second scenario is supported by an mtDNA study that inferred major influxes of Papuan ancestry into Wallacea: after the Last Glacial Maximum (around 15 ka); and Austronesian contact (around 3 ka)[45].

Previous studies also reported elevated amounts of Denisovan ancestry in present-day Wallaceans, which correlate with the amount of Papuan-related ancestry[46]. We confirmed that the same relationship holds for ancient Wallaceans (Extended Data Fig. 5); therefore, their Denisovan-related ancestry was probably contributed via Papuan-related admixture.

**Early SEA ancestry in NTT.** Wallacea is generally assumed to have been mostly isolated and shaped by two main streams of ancestry: one related to the first settlement of Sahul and another associated with the Austronesian expansion[28–30]. The results presented in this study show that the genetic variation of ancient individuals from NTT also requires ancestry contributions from MSEA. The inferred order of events makes it unlikely that the SEA and Austronesian-related ancestries were introduced together from Western Indonesia, where both ancestries are found. Instead, it seems that human groups from MSEA crossed into southern Wallacea before the Austronesian-related groups spread into the region. Further support for this scenario comes from genetic analyses of the commensal black rat—often a good indicator of human migration—which suggests that this species was also first introduced in NTT from MSEA[47].

The broad geographical distribution of groups best matching the MSEA ancestry of southern Wallaceans raises questions about the actual origin(s) of peoples who reached those islands.

The best present-day proxies, the Mlabri from Thailand/Laos and the Nicobarese from the Nicobar islands, speak Austroasiatic languages and have been relatively isolated compared to other MSEA groups that recently experienced extensive admixture[48–51]. Their isolation might explain why they appear as best proxies without being necessarily connected to the inferred migration event. Moreover, there is no clear link between any specific ancient group from MSEA and the actual source that contributed ancestry to NTT since we identified several equivalent ancient proxies for this ancestry.

Other evidence for this migration is equivocal. NTT languages are either Austronesian or Papuan-related and no influences from MSEA language families have been reported. Similarly, there is no archaeological evidence for pre-Austronesian contact between MSEA and southern Wallacea; the earliest evidence is the appearance of the Đông Sơn bronze drums, which spread to southern but not northern Wallacea around the early centuries AD, following maritime trade routes[52]. These drums probably originated in northern Vietnam or adjacent provinces of southern China[52]. Although we cannot rule out some MSEA ancestry contributions from the Đông Sơn period (or even later) for the younger NTT individuals, the high amount of MSEA ancestry in the Liang Bua individual (2,600 BP) and our AHG inferences support an earlier presence in southern Wallacea. This has important implications for our understanding of the Neolithic expansion into ISEA since it brings to light a previously undescribed human dispersal from MSEA. Future archaeological studies in SEA might help link this human dispersal to any contemporaneous material culture. Additionally, ancient DNA from older periods will help clarify the time of arrival of this ancestry.

**Austronesian expansion into the North Moluccas and Pacific.** The fine-scale structure observed among Austronesian-related groups from ISEA and the Pacific, and the higher genetic proximity of the ancient North Moluccans to the latter, are pertinent for previous considerations of the role of the North Moluccas in dispersals to the Pacific[20]. When analysed through seafaring and climatic models, the North Moluccas is one of the most likely starting points for settlers that ventured into the Palau or Mariana Islands (western Micronesia)[53,54]. Their geographical setting also led archaeologists to search the region for pottery that might be ancestral to the Lapita cultural complex (distributed from the Bismarck Archipelago to Samoa), as well as the Marianas Redware culture[11,18]. However, current evidence does not connect the North Moluccas red-slipped pottery to either of these material cultures but instead to pottery from the Talaud Islands, northern and western Sulawesi, North Luzon, Batanes and south-eastern Taiwan[11].

The genetic affinity between the ancient individuals from the North Moluccas and the Mariana Islands suggested by our results has some parallels in mtDNA studies based on present-day groups[55]. However, ancient DNA from Guam supports an origin for the settlement of the Mariana Islands from the Philippines[39]. Under a simple expansion scenario, without back migration, the increasing amounts of Austronesian ancestry characteristic of the Pacific (and decrease of ancestry characteristic of Taiwan/Philippines) from the ancient North Moluccas to Guam (2,200 BP), Vanuatu (2,900 BP) and Tonga (2,500 BP) could reflect their relative position along the peopling wave that eventually reached the eastern parts of the Pacific (Extended Data Fig. 3). Yet, the position in this study refers to the split order of groups without any necessary attachment to their geographical location. Therefore, it is possible that the higher proximity between the North Moluccas and groups from the Pacific, compared to NTT or Sulawesi, simply reflects their more recent ancestry tracing back to a common Austronesian source, regardless of its location. This scenario also implies that the Austronesian-related ancestry found in NTT or Sulawesi is somewhat differentiated from that found in the North Moluccas. Nonetheless, we cannot exclude the possibility of more complex migration scenarios (for example, involving back migrations).

It is also important to consider that the dates of the oldest individuals from the North Moluccas (2,150 BP in Morotai and 1,900 BP in Kayoa Island) overlap with the start of the Early Metal Age 2,300–2,000 ya in this region[11]. This period is characterized by the appearance of copper, bronze and iron artefacts and glass beads in the region, as well as the spread of pottery into Morotai. Thus, these individuals might not be good representatives of the first Austronesians, thought to have reached Kayoa island 3,500 ya[11,18], but instead might reflect additional genetic influences brought by later contacts.

However, linguistic evidence parallels the genetic evidence for a closer relationship of North Moluccans with Oceanians, compared to peoples from NTT. The Austronesian (Malayo-Polynesian major subgroup) languages of the Northern Moluccas are part of the South Halmahera–West New Guinea (SHWNG) regional subgroup, which are closer to Oceanic languages than to any other Western Malayo-Polynesian major subgroup[13,56,57], whereas the languages spoken in NTT are an outgroup to both SHWNG and Oceanic languages[13].

**The timing of admixture.** Besides providing direct evidence for Austronesian-Papuan contact before 2,150 BP in the North Moluccas and 2,600 BP in NTT, the oldest individuals gave admixture date estimates close to 3,000 BP. This period is slightly younger than the earliest archaeological traces of the Neolithic (Austronesian) arrival in the North Moluccas (approximately 3,500 BP for Kayoa Island[11,18]) but predates the adoption of pottery on Morotai Island (2,300–2,000 BP), where the oldest North Moluccan individuals in this study were found[11,58]. However, it is similar to some of the earliest secure dates from NTT (3,000 BP for eastern Flores)[17]. Previous studies conducted on present-day eastern Indonesian populations suggested that this admixture lagged about a millennium behind the arrival of Austronesian populations[30]. Our admixture analysis for ancient individuals, and the comparison with present-day data, provides an alternative explanation and helps to clarify previous debates concerning admixture times[28–30,32,59]. The decreasing trend in admixture time estimates from the oldest individuals until present-day populations is a strong indicator of multiple pulses or continuous admixture. Therefore, even our oldest estimates might not correspond to the actual start of admixture but to a more recent time due to additional gene flow.

Gene flow events might have been facilitated by emergent maritime networks and spice trade interactions in the Metal Age[11]. In the North Moluccas, this period not only corresponds to a more

rapid spread of material culture between regions[11,21,22,58] but also to the period of language levelling or radiation described for both the Austronesian (SHWNG) and Papuan (West Papuan phylum, Northern Halmahera stock) languages[11]. The historical socio-economic systems of the North Moluccas and western Papua also brought together Papuan-speaking resident populations and a Malay-speaking elite[11], thus mixing could have occurred until very recently. In contrast to the North Moluccas, NTT and Sulawesi individuals do not show genetic traces of very recent contact. Still, their demographic history was nonetheless characterized by a long-term process of admixture involving at least two Asian-related ancestries.

The evidence for ongoing contact in Wallacea has important implications for efforts that use present-day genomic data to discern the direction and number of human migrations to Sahul (for example, Brucato et al.[60]). Failure to consider such contact may result in wrongly considering the genetic affinity between Papuans and northern versus southern Wallaceans to reflect ancestral relationships of these groups rather than differences in the degree of contact. Overall, our findings suggest different histories for northern versus southern Wallaceans that reflect differences in contact with MSEA, in the duration of contact with Papuans and perhaps even with different Austronesian-related groups. Future ancient DNA studies involving individuals from earlier periods will help to improve our understanding of the demographic changes occurring before and after the arrival of Austronesians in Wallacea.

## Methods

**Sampling.** All samples were processed in dedicated ancient DNA laboratories at the Max Planck Institute for the Science of Human History and the Max Planck Institute for Evolutionary Anthropology. At the Max Planck Institute for the Science of Human History, the petrous bone of samples AMA001, AMA004 and AMA009 was first drilled from the outside, identifying the position of the densest part by orienting on the internal acoustic metre and drilling parallel to it into the target area to avoid damaging the semicircular ducts[61] (protocol: https://doi.org/10.17504/protocols.io.bqd8ms9w). After that, the petrous bone was cut along the margo superior partis petrosae (crista pyramidis) and 50–150 mg of bone powder were drilled from the densest part around the cochlea[62]. All other elements processed at the Max Planck Institute for the Science of Human History (AMA003, AMA005, AMA008, JAB001, KMO001, LIA001, LIA002, LIT001, TOP002, TOP004) were sampled by cutting and drilling the densest part. At the Max Planck Institute for Evolutionary Anthropology, the Tanjung Pinang and Uattamdi specimens were sampled by targeting the cochlea from the outside. For this, a thin layer of surface (approximately 1 mm) was removed with a sterile dentistry drill. Small holes were then drilled into the cleaned areas, yielding between 42 and 63 mg of bone powder. Detailed information on the analysed samples, radiocarbon dating and archaeological context are provided in the supplementary information and in Supplementary Tables 2 and 12.

**DNA extraction.** DNA extraction in both laboratories was carried out using a silica-based method optimized for the recovery of highly degraded DNA[63,64]. To release DNA from 50–100 mg of bone powder, a solution of 900 μl EDTA, 75 μl $H_2O$ and 25 μl proteinase K was added. In a rotator, samples were digested for at least 16 h at 37 °C, followed by an additional hour at 56 °C. The suspension was then centrifuged and transferred into a binding buffer. To bind DNA, large-volume silica spin columns (High Pure Viral Nucleic Acid Large Volume Kit; Roche Molecular Systems) were used. After two washing steps using the manufacturer's wash buffer, DNA was eluted in TET (10 mM Tris, 1 mM EDTA and 0.05% Tween 20). At the Max Planck Institute for the Science of Human History, the second elution of DNA from the spin column was carried out using a fresh aliquot of elution buffer for a total of 100 μl DNA extract, whereas at the Max Planck Institute for Evolutionary Anthropology the same aliquot of elution buffer was loaded twice for a total of 50 μl DNA extract (protocol: https://doi.org/10.17504/protocols.io.baksicwe).

**Library preparation.** At the Max Planck Institute for the Science of Human History, double-stranded DNA libraries were built from 25 μl of DNA extract in the presence of uracil DNA glycosylase (UDG) (half libraries) according to a protocol that uses the UDG enzyme to reduce, but not eliminate, the amount of deamination-induced damage towards the ends of ancient DNA fragments[65]. Negative and positive controls were carried alongside each experiment (extraction and library preparation) (protocol: https://doi.org/10.17504/protocols.io.bmh6k39e). Libraries were quantified with the IS7 and IS8 primers[66] in a quantification assay using a DyNAmo SYBR Green qPCR Kit (Thermo Fisher Scientific) on the LightCycler 480 (Roche). Each ancient DNA library was double-indexed[67] in parallel 100 μl reactions using PfuTurbo DNA Polymerase

(Agilent Technologies) (protocol: https://doi.org/10.17504/protocols.io.bakticwn). The indexed products for each library were pooled, purified over MinElute columns (QIAGEN), eluted in 50 μl TET and again quantified with the IS5 and IS6 primers[66] using the quantification method described above; 4 μl of the purified product were amplified in multiple 100 μl reactions using Herculase II Fusion DNA Polymerase (Agilent Technologies) according to the manufacturer's specifications with 0.3 μM of the IS5/IS6 primers. After another MinElute purification, the product was quantified with the Agilent 2100 Bioanalyzer DNA 1000 chip. An equimolar pool of all libraries was then prepared for shotgun sequencing on the Illumina HiSeq 4000 platform using an SR75 sequencing kit. Libraries were further amplified with IS5/IS6 primers to reach a concentration of 200–400 ng μl⁻¹ as measured on a NanoDrop spectrophotometer (Thermo Fisher Scientific). At the Max Planck Institute for Evolutionary Anthropology, single-stranded DNA libraries were prepared without UDG treatment using the Bravo NGS Workstation B (Agilent Technologies), exactly as described in Gansauge et al.[68]. Briefly, after an initial denaturation step, adaptor oligonucleotides are ligated to the 3′ ends of the single-stranded ancient DNA fragments using T4 DNA ligase. Using streptavidin-covered magnetic beads, the ligation products and excess adaptors were immobilized, a primer hybridized to the adaptor and a copy of the ancient DNA molecule generated using the Klenow fragment of *Escherichia coli* DNA polymerase I. Excess primer was then removed in a washing step at increased temperature, which prevented the formation of adaptor dimers. Blunt-end ligation with T4 DNA ligase was used to ligate a second, double-stranded adaptor. Finally, the library strand was released from the beads by heat denaturation. Libraries were quantified through two probe-based quantitative PCR assays and amplified and indexed via PCR[68].

**Targeted enrichment and high-throughput sequencing.** MtDNA capture[69] was performed on screened libraries which, after shotgun sequencing, showed the presence of ancient DNA, highlighted by the typical C to T and G to A substitution pattern towards the 5′ and 3′ molecule ends, respectively. Furthermore, samples with a percentage of human DNA in shotgun data around 0.1% or greater were enriched for a set of 1,237,207 targeted SNPs across the human genome (1,240 K capture)[33]. The enriched DNA product was sequenced on an Illumina HiSeq 4000 instrument with 75 cycles single reads or 50 cycles paired-end reads according to the manufacturer's protocol (at the Max Planck Institute for the Science of Human History) or on a HiSeq 2500 with 75 paired-end reads (at the Max Planck Institute for Evolutionary Anthropology). The output was demultiplexed using in-house scripts requiring either a perfect match of the expected and observed index sequences (Max Planck Institute for Evolutionary Anthropology samples) or allowing a single mismatch between the expected and observed index sequences (Max Planck Institute for the Science of Human History samples).

**Genomic data processing.** Preprocessing of the sequenced reads was performed using EAGER v.1.92.55 (ref. [70]). The resulting reads were clipped to remove residual adaptor sequences using Clip&Merge v.1.7.6[71] and AdapterRemoval v.2 (ref. [72]). Clipped sequences were then mapped against the human reference genome hg19 using the Burrows–Wheeler Aligner v.0.7.12 (ref. [73]), disabling seeding (-l 16,500) and allowing for 2 mismatches (--n 0.01). Duplicates were removed with DeDup v.0.12.2 (ref. [70]). Additionally, a mapping quality filter of 30 was applied using SAMtools v.1.3 (ref. [74]). Different sequencing runs and libraries from the same individuals were merged and duplicates removed and sorted again using SAMtools v.1.3 (ref. [74]). Genotype calling was performed separately for trimmed and untrimmed reads using pileupCaller v.8.6.5 (https://github.com/stschiff/sequenceTools), a tool that randomly draws one allele at each of the targeted SNPs covered at least once. For the UDG-treated libraries produced at the Max Planck Institute for the Science of Human History, two bases were trimmed on both ends of the reads. For libraries produced at the Max Planck Institute for Evolutionary Anthropology (without UDG treatment), the damage plots were inspected to determine the number of bases to trim from each read. For all libraries, the residual damage extended 8 base pairs into the read, after which it was below 0.05%, and trimmed accordingly. We combined the genotypes keeping all transversions from the untrimmed genotypes and transitions only from the trimmed genotypes to eliminate problematic, damage-related transitions overrepresented at the ends of reads. The generated pseudo-haploid calls were merged with previously published ancient data[38,39,42,48,75–79], present-day genomes from the Simons Genome Diversity Project[80], and worldwide populations genotyped on the Affymetrix Human Origins array[38,41,50,75,76,81–86]. For the PCA and DyStruct analyses, we additionally merged the data with populations from ISEA genotyped on the Affymetrix 6.0 (refs. [46,87]) (dataset 1) or Affymetrix Axiom Genome-Wide Human Array[30] (dataset 2), filtering out SNPs with a missing rate higher than 10%. Related individuals were excluded if they exhibited a proportion of identity by descent (IBD) higher than 0.3, computed in PLINK v.1.9 (ref. [88]) as P(IBD = 2) + 0.5 × P(IBD = 1). We additionally pruned datasets 1 and 2 for linkage disequilibrium with PLINK v.1.9, removing SNPs with r² > 0.4 in 200 kilobase windows, shifted at 25-SNP intervals. After pruning, a total of 89,597 and 65,880 SNPs remained in datasets 1 and 2, respectively.

Y-chromosome haplogroups were identified by calling the SNPs covered on the Y chromosome of all male individuals using the pileup from the Rsamtools v1.3.[89] package and by recording the number and form of derived and ancestral SNPs

overlapping with the International Society of Genetic Genealogy SNP index v.14.07 (https://github.com/Integrative-Transcriptomics/DamageProfiler)[90].

**Authentication of ancient DNA.** The typical features of ancient DNA were inspected with DamageProfiler v.0.3.1 (http://bintray.com/apeltzer/EAGER/DamageProfiler)[70]. Sex determination was performed by comparing the coverage on the targeted X-chromosome SNPs to the coverage on the Y-chromosome SNPs, both normalized by the coverage on the autosomal SNPs[71] (Supplementary Table 2). For male individuals, ANGSD v.0.919 was run to measure the rate of heterozygosity of polymorphic sites on the X chromosome after accounting for sequencing errors in the flanking regions[91]. This provides an estimate of nuclear DNA contamination in males since they are expected to have only one allele at each site. For both male and female individuals, mtDNA-captured data were used to jointly reconstruct the mtDNA consensus sequence and estimate contamination levels with contamMix v.1.0-10[69] (Supplementary Table 2) using an in-house pipeline (https://github.com/alexhbnr-mitoBench-ancientMT[39]).

**Statistical analyses.** PCAs were carried out using smartpca v.10210 (ref. [92]) based on present-day Asian and Oceanian populations from datasets 1 and 2. Ancient individuals were projected onto the calculated components using the options lsqproject: YES and numoutlieriter: 0. We used DyStruct v.1.1.0 (ref. [40]) to infer shared genetic ancestry taking into account archaeological age. The uncalibrated radiocarbon dates of each ancient sample were converted to generations, assuming a generation time of 29 years[93]. For each dataset (1 and 2), we performed 25 independent runs, using 2–15 ancestral populations (*K*). To compare runs for different values of *K*, a subset of loci (5%) was held out during training and the conditional log-likelihood was subsequently evaluated (Supplementary Fig. 1a,b). Within the best *K*, the run with the highest objective function was selected (Supplementary Fig. 1c,f).

To formally test population relationships we used the $f_4$-statistics implemented in the ADMIXTOOLS software v.4.1[41]. This analysis was carried out using the admixr v.0.9.1 R package[94]. To evaluate differences in $f_4$-statistics for individuals from NTT and the North Moluccas, we built a Bayesian linear regression model:

$$B_{\text{OBS},i} \sim \text{Normal}\,(B_{\text{TRUE},i}, B_{\text{SE},i})$$

$$B_{\text{TRUE},i} \sim \text{Normal}(\mu_i, \sigma)$$

$$\mu_i = \alpha_{\text{REGION}[i]} + \beta_{\text{REGION}[i]}A_{\text{TRUE},i}$$

$$A_{\text{OBS},i} \sim \text{Normal}\,(A_{\text{TRUE},i}, A_{\text{SE},i})$$

$$A_{\text{TRUE},i} \sim \text{Normal}\,(0, 1)$$

$$\alpha_{\text{REGION}[i]} \sim \text{Normal}\,(0, 1)$$

$$\beta_{\text{REGION}[i]} \sim \text{Normal}\,(0, 10)$$

$$\sigma \sim \text{Exponential}\,(1)$$

The model was stratified by region (NTT versus North Moluccas) and takes into account measurement error in both $A_{\text{OBS}}$ and $B_{\text{OBS}}$ variables (corresponding to the $f_4$-statistics displayed on the x and y axes of the biplots, respectively)[95]. The parameters $\mu$ and $\sigma$ represent the mean and s.d. $A_{\text{OBS}}$ and $B_{\text{TRUE}}$ correspond to the unobserved true values of A and B. Both variables were standardized. The posterior distribution was obtained via Hamiltonian Monte Carlo approximation as implemented in the R package rethinking v2.21 (https://github.com/rmcelreath/rethinking), using 6 chains of 4,000 samples. We used a non-centred parameterization of the error model to aid in posterior exploration. Convergence of the chains was assessed by inspection of the trace plots, Rhat and the effective number of samples. All of these criteria indicate reliable sampling. All Rhat values were equal to 1.00 and all effective number of samples values were above 500. This procedure was applied to each pair of $f_4$-statistics separately. The code is available at https://github.com/sroliveiraa/ancient_Wallacea_f4_differences.

We used qpWave v.410 (ref. [96]) and qpAdm v.650 (ref. [41]) to test two- and three-wave admixture models, using a 'rotating' strategy[97]. A reference set of populations was chosen to represent diverse human groups and include potential source populations for the ancient Wallacean individuals: Mbuti, English, Brahui, Onge, Yakut, Oroqen, Lahu, Miao, Dai, Khomu, Denisova, Papuan, Kankanaey and Mlabri. We rejected models if their P values were lower than 0.01, if there were negative admixture proportions or if the s.e. was larger than the corresponding admixture proportion. When more than one model was accepted (Supplementary Table 7), the estimated admixture proportions under the model with the highest P value was preferred and used in subsequent analyses (Supplementary Fig. 10 and Extended Data Fig. 5) because the results better matched the DyStruct

ancestry proportions and the ability to reject models might be affected by several factors (for example, the ancestry proportion, the quality of the target sample, the combination of ancient and present-day samples in the same analysis). The correlation between ancestry proportions inferred with qpAdm and DyStruct was assessed with a Mantel test with 10,000 permutations of the distance matrix to determine significance.

The relative order of the mixing of different ancestries was inferred using the AHG approach[39]. Assuming that an admixed population with two ancestry components (A and B) later receives a third component (C) via admixture, then ancestry components A and B from the first admixture event will covary with the component that comes later (C) but the ratio of A and B throughout the population will be independent from C. Thus, the covariance of the recent ancestry C with the ratio of the two older ancestries A and B should be zero. The AHG approach thus involves estimating the covariance of the frequencies of A/B with C, A/C with B and B/C with A across all individuals in the population; the covariance closest to 0 then indicates the order of admixture events. We used the DyStruct ancestry proportions for each ancient and present-day Wallacean individual included in dataset 1 and 2 (Supplementary Tables 8 and 9) to calculate the covariances between ancestry components as indicated in Supplementary Table 10. The sequence of admixture events was then determined by the configuration that produced the smallest absolute value of the covariance estimate. The time since admixture was estimated based on the decay of ancestry covariance using the software DATES v.753 (ref. [44]) with the following parameters: binsize = 0.001; maxdis = 1.0; jackknife, YES; qbin = 10; runfit, YES; afffit, YES; lovalfit = 0.45; mincount = 1. In our main analysis, to maximize the number of SNPs included in the analysis and have equal sample sizes, we used as sources 16 Papuan individuals and 16 Asian-related individuals (2 Amis, 1 Atayal, 2 Kankanaey, 5 Dai, 2 Dusun, 2 She, 2 Kinh) with data covering the approximate 1,240,000 SNPs captured in the ancient samples. Two additional admixture tests were conducted using the same Papuan source and either Austronesians (2 Amis, 1 Atayal, 2 Kankanaey) or MSEA (2 Cambodian, 2 Kinh, 2 Thai) as the Asian-related source.

**Reporting summary.** Further information on research design is available in the Nature Research Reporting Summary linked to this article.

## Data availability

All newly reported ancient DNA data, including nuclear DNA and mtDNA alignment sequences, are archived in the European Nucleotide Archive (accession no. PRJEB48109).

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

## Acknowledgements

We thank R. Radzeviciute, A. Wissgott, the Max Planck Institute for Evolutionary Anthropology lab technicians and the Max Planck Institute for Evolutionary Anthropology Sequencing and Bioinformatics Groups for their excellent support, C. Jeong for valuable comments, L. Iasi for helpful discussions on admixture dating and R. McElreath for help with statistics. This research was supported by the Max Planck Society. K.N., S.C. and A.P. were supported by the European Research Council Starting Grant 'Waves' (no. ERC758967). The research conducted on the samples from Liang Toge, Liang Bua and Komodo was part of a New Zealand Fast-Start Marsden Grant (no. 18-UOO-135). The research conducted on the Jareng Bori site was part of a joint project between the Australian National University Universitas Gadjah Maja funded by an ARC Laureate Project no. FL120100156.

## Author contributions

R.O., P.B., R.K., T.K. and S.H. contributed archaeological material, collected with the critical support of A.A.O., M.T., C.K., D.B.M., R.S.P., M., A.P. and S.O.C. T.H., C.F.W.H., R.K., F.P. and P.B. contributed the radiocarbon data. K.N., S.C. and M.M. conducted the ancient DNA laboratory work. S.O., K.N., S.C., I.P. and A.H. performed the genetic analysis. S.O. and K.N. wrote the manuscript with input from all authors. M.S., C.P. and J.K. conceived and coordinated the study.

## Funding

## Competing interests

The authors declare no competing interests.

## Additional information

**Extended data** is available for this paper at https://doi.org/10.1038/s41559-022-01775-2.

**Correspondence and requests for materials** should be addressed to Sandra Oliveira, Johannes Krause, Cosimo Posth or Mark Stoneking.

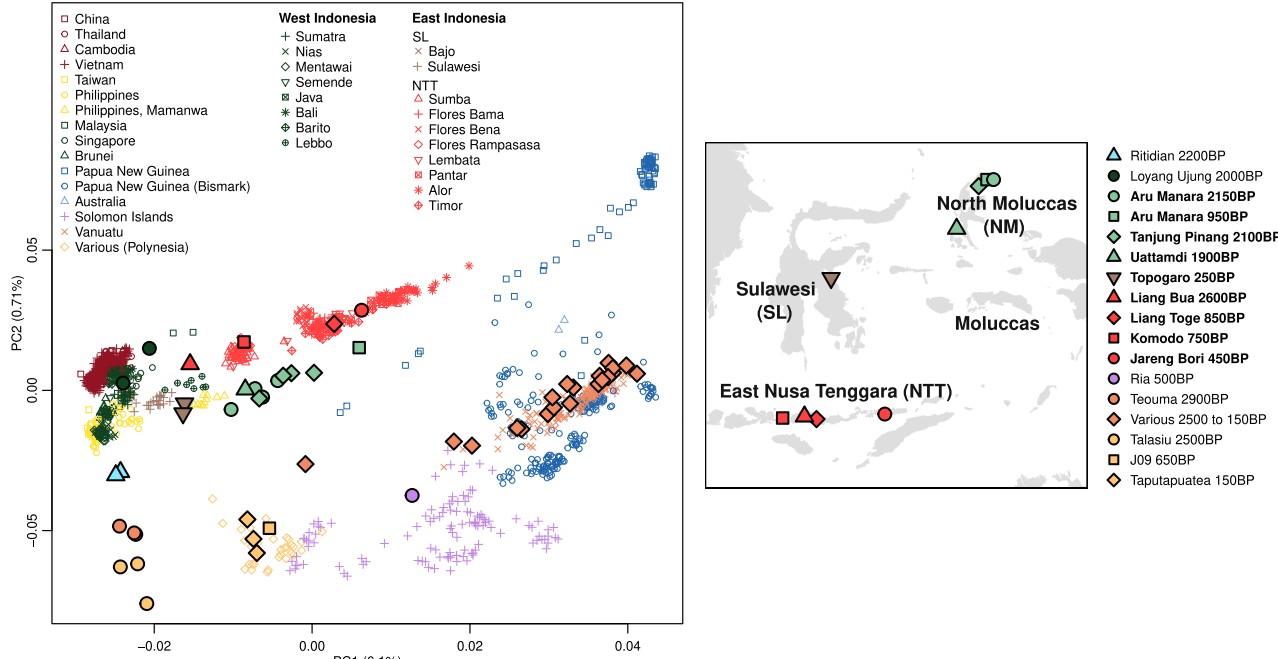

**Extended Data Fig. 1 | PCA of publicly available whole genome data merged with Human Origins genotype data and Affymetrix Axiom Genome-Wide Human genotype data (dataset 2).** Ancient individuals (shown with a black contour) are projected and their fill color matches the color of present-day samples from the same geographic area. The location of the ancient individuals newly presented in this study is shown in the map on the right panel. For our purposes East Nusa Tenggara is abbreviated to NTT.

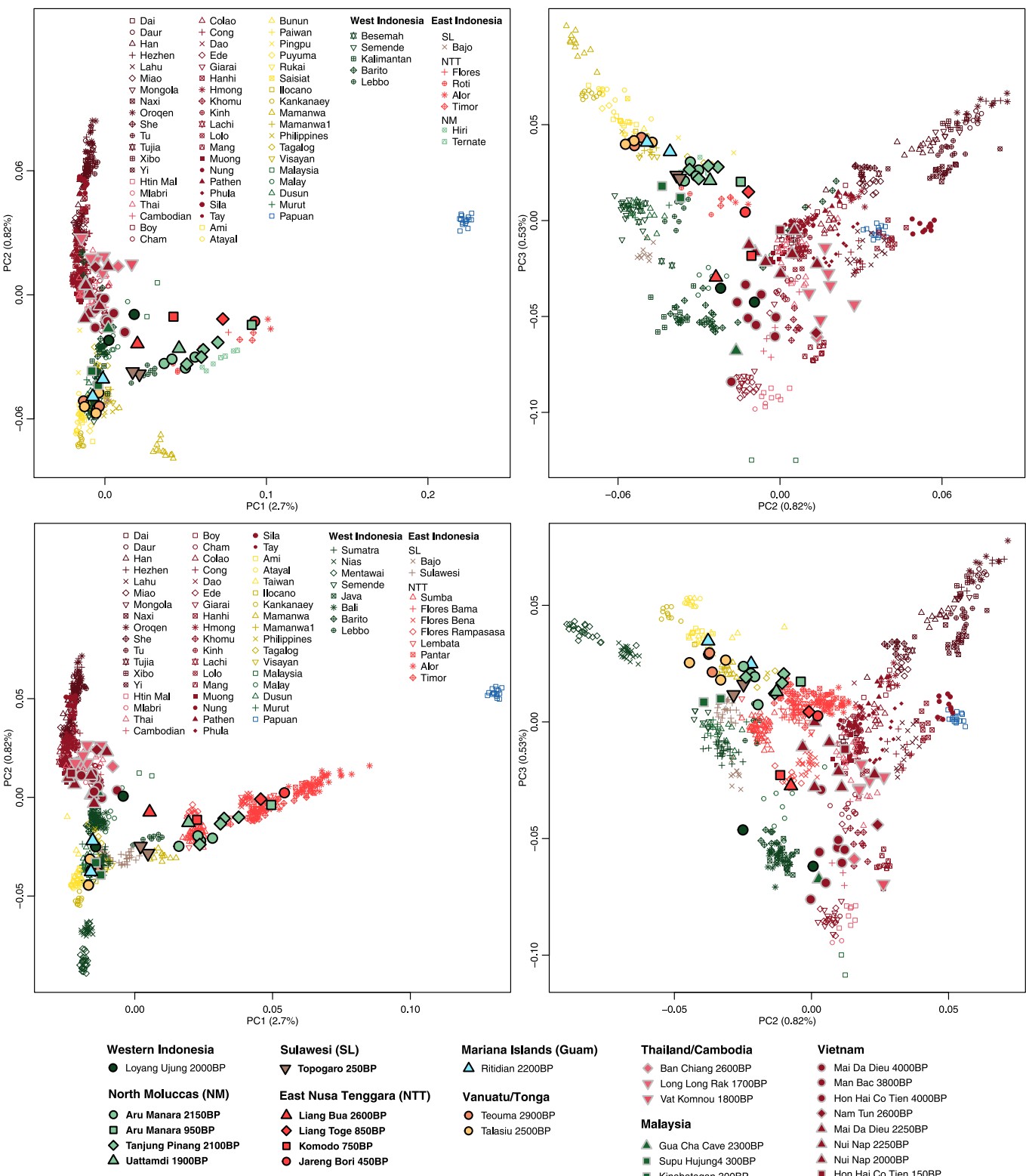

**Extended Data Fig. 2 | PCA generated with a subset of present-day populations from dataset 1 (A) and dataset 2 (B) that emphasize differences between closely related Asian ancestries.** Ancient individuals (shown with a black/grey contour) are projected and their fill color matches the color of present-day samples from the same geographic area. For our purposes East Nusa Tenggara is abbreviated to NTT.

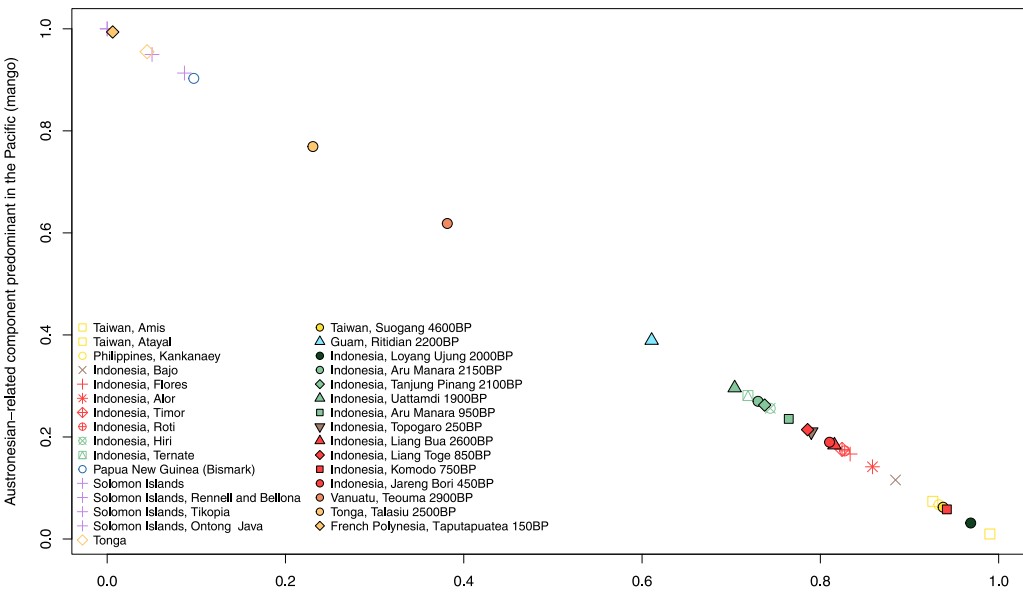

**Extended Data Fig. 3 | Representation of two Austronesian-related components.** The frequencies of the "yellow" and "mango" components identified in Supplementary Figure 1E were normalized to sum to 1.

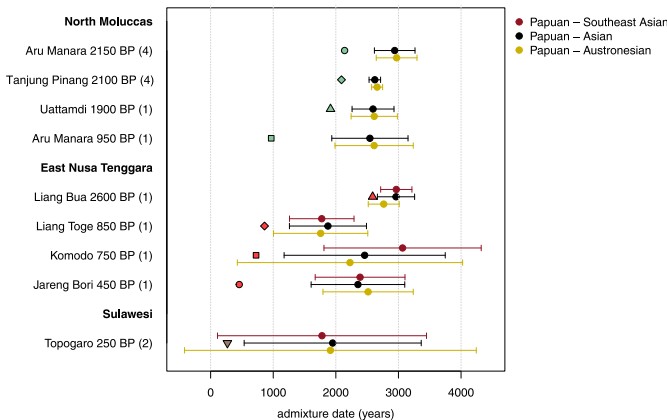

**Extended Data Fig. 4 | Admixture dates estimated with different source groups.** The admixture dates estimated with a pool of Asian groups are shown in black, while the admixture dates estimated with a more specific set of Asian-related groups (SEA or Austronesians) are shown in dark red and yellow. Data are presented as point estimates ± 2 SE. The individuals age is shown by filled symbols with a black contour. The number of individuals included in each group is shown in parenthesis, next to the group label.

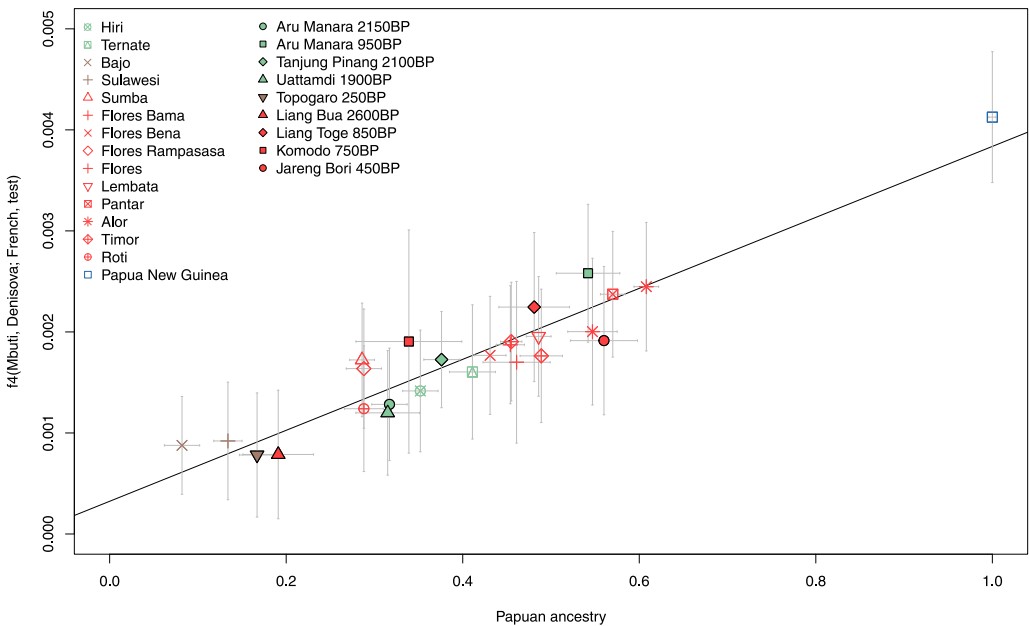

**Extended Data Fig. 5 | Papuan vs Denisova ancestry.** The x-axis presents the Papuan-related ancestry proportion ± 2 SE calculated with block jackknife in the qpAdm software. The Denisova ancestry is represented in the y-axis by an $f_4$-statistic of the form $f_4$(*Mbuti, Denisova; French, test*) ± 2 SE.

# Reporting Summary

## Statistics

For all statistical analyses, confirm that the following items are present in the figure legend, table legend, main text, or Methods section.

| n/a | Confirmed | |
|---|---|---|
| ☐ | ☒ | The exact sample size (*n*) for each experimental group/condition, given as a discrete number and unit of measurement |
| ☐ | ☒ | A statement on whether measurements were taken from distinct samples or whether the same sample was measured repeatedly |
| ☐ | ☒ | The statistical test(s) used AND whether they are one- or two-sided<br>*Only common tests should be described solely by name; describe more complex techniques in the Methods section.* |
| ☐ | ☒ | A description of all covariates tested |
| ☐ | ☒ | A description of any assumptions or corrections, such as tests of normality and adjustment for multiple comparisons |
| ☐ | ☒ | A full description of the statistical parameters including central tendency (e.g. means) or other basic estimates (e.g. regression coefficient) AND variation (e.g. standard deviation) or associated estimates of uncertainty (e.g. confidence intervals) |
| ☐ | ☒ | For null hypothesis testing, the test statistic (e.g. *F*, *t*, *r*) with confidence intervals, effect sizes, degrees of freedom and *P* value noted<br>*Give P values as exact values whenever suitable.* |
| ☐ | ☒ | For Bayesian analysis, information on the choice of priors and Markov chain Monte Carlo settings |
| ☒ | ☐ | For hierarchical and complex designs, identification of the appropriate level for tests and full reporting of outcomes |
| ☐ | ☒ | Estimates of effect sizes (e.g. Cohen's *d*, Pearson's *r*), indicating how they were calculated |

*Our web collection on statistics for biologists contains articles on many of the points above.*

## Software and code

Policy information about availability of computer code

| Data collection | No software was used for data collection |
|---|---|

| Data analysis | EAGER v.1.92.55<br>ClipAndMerge<br>AdapterRemoval v.2<br>BWA v.0.7.12<br>DeDup v.0.12.2<br>SAMtools v1.3<br>pileupCaller v.8.6.5<br>EIGENSOFT v.7.2.1 (mergeit, convertf)<br>DamageProfiler v.0.3.1<br>ANGSD v.0.919<br>contamMix<br>PLINK v.1.9<br>smartpca v.10210<br>DyStruct v.1.1.0<br>ADMIXTOOLS v.4.1<br>admixr R package<br>rethinking R package<br>DATES v.753<br>RStudio v.1.1.38<br>OxCal v.4.4 |
|---|---|

For manuscripts utilizing custom algorithms or software that are central to the research but not yet described in published literature, software must be made available to editors and reviewers. We strongly encourage code deposition in a community repository (e.g. GitHub). See the Nature Portfolio guidelines for submitting code & software for further information.

## Data

Policy information about availability of data

All manuscripts must include a data availability statement. This statement should provide the following information, where applicable:
- Accession codes, unique identifiers, or web links for publicly available datasets
- A description of any restrictions on data availability
- For clinical datasets or third party data, please ensure that the statement adheres to our policy

Alignment files of the nuclear and mitochondrial DNA sequences for the newly sequenced individuals are available at the ENA database under the accession number PRJEB48109.

## Field-specific reporting

Please select the one below that is the best fit for your research. If you are not sure, read the appropriate sections before making your selection.

☒ Life sciences          ☐ Behavioural & social sciences          ☐ Ecological, evolutionary & environmental sciences

For a reference copy of the document with all sections, see nature.com/documents/nr-reporting-summary-flat.pdf

## Life sciences study design

All studies must disclose on these points even when the disclosure is negative.

| Sample size | Sample size was not predetermined and was based on the availability of skeletal material. |
|---|---|
| Data exclusions | No data from the newly generated ancient samples were excluded. Related individuals, identified among the previously published datasets used to contextualize genetic variation, were excluded according to a pre-defined criteria outlined in the methods. |
| Replication | Principal component, DyStruct, qpAdm, Admixture History Graph (AHG), and admixture dating analyses were carried out based on two different datasets that included present-day populations from Island Southeast Asia genotyped on the Affymetrix 6.0 (dataset 1) and the Affymetrix Axiom Genome-Wide Human Array (dataset 2). The findings were consistent for all analyses. |
| Randomization | We grouped ancient samples from the same archaeological site based on radiocarbon dates or archaeological context when relevant to improve the power of estimations, and grouped present-day samples according to the population of origin. We confirmed that samples from the same site and age have a similar genetic profile in unsupervised analysis (PCA and DyStruct) before grouping them. |
| Blinding | Blinding was not relevant for this study. The data analysis was performed for all individuals separately or into subgroups defined by external information (archaeological context and dates). |

## Reporting for specific materials, systems and methods

We require information from authors about some types of materials, experimental systems and methods used in many studies. Here, indicate whether each material, system or method listed is relevant to your study. If you are not sure if a list item applies to your research, read the appropriate section before selecting a response.

## Materials & experimental systems

| n/a | Involved in the study |
|-----|----------------------|
| ☒ | ☐ Antibodies |
| ☒ | ☐ Eukaryotic cell lines |
| ☐ | ☒ Palaeontology and archaeology |
| ☒ | ☐ Animals and other organisms |
| ☒ | ☐ Human research participants |
| ☒ | ☐ Clinical data |
| ☒ | ☐ Dual use research of concern |

## Methods

| n/a | Involved in the study |
|-----|----------------------|
| ☒ | ☐ ChIP-seq |
| ☒ | ☐ Flow cytometry |
| ☒ | ☐ MRI-based neuroimaging |

# Palaeontology and Archaeology

**Specimen provenance**
The specimens reported in this study come from eight archaeological sites in Indonesia: Aru Manara, Tanjung Pinang, Gua Uattamdi, Topogaro, Liang Bua, Liang Toge, Komodo, and Jareng Bori. The archaeological context and detailed information on the specimens collection are described in the Supplementary information.

**Specimen deposition**
The remaining portion of the bones sampled for DNA analysis from the Aru Manara, Topogaro, Tanjung Pinang, and Uattamdi sites will be deposited at the Pusat Penelitian Arkeologi Nasional (Jakarta, Indonesia), and those from Jareng Bori, Komodo, Liang Bua, and Liang Toge will be returned to Prof. Koesbardiati at Universitas Airlangga (Surabaya, Indonesia).

**Dating methods**
For this study we obtained 13 direct radiocarbon dates. Five radiocarbon dating laboratories were used: Curt-Engelhorn-Zentrum Archäometrie gGmbH in Mannheim (Germany), Radiocarbon Dating Laboratory at the University of Waikato (New Zealand), the University of Oxford's Radiocarbon Accelerator Unit (UK), the Australian National University in Canberra (Australia), and the Iso-trace Research Department of Chemistry, in the University of Otago (New Zealand). Pretreatment processes, quality control protocols, and dating methods used by each laboratory are provided in Supplementary Information and Supplementary Table 12. Conventional radiocarbon ages were calibrated using the OxCal 4.4 program and the INTCAL20 calibration curve, with uncertainties reported at 68% and 95% confidence interval.

☒ Tick this box to confirm that the raw and calibrated dates are available in the paper or in Supplementary Information.

**Ethics oversight**
For the skeletal material from Tanjung Pinang and Uattamdi, the research was undertaken as part of a collaborative project between Pusat Penelitian Arkeologi Nasional (Jakarta) and the Australian National University, under Lembaga Ilmu Pengetahuan Indonesia Research Permits 6939/S.K./1990, 307/I/KS/1994, and 10290/V3/KS/1995. For the skeletal material from Aru Manara, the research was undertaken as part of a collaborative project between Pusat Penelitian Arkeologi Nasional (Jakarta) and the Tokai University, under Kementarian Riset dan Teknologi Research Permits 0291/SIP/FRP/VIII/2011 and Pusat Penelitian Arkeologi Nasional export permit UM.001/2595/PAN/KPK/IX/2012, 2634/H5/TU/2017. For the skeletal material from Topogaro, the research was undertaken as part of a collaborative project between Pusat Penelitian Arkeologi Nasional (Jakarta) and the Tokai University, under Kementarian Riset dan Teknologi Research Permits 40/EXT/SIP/FRP/SM/VII/2015 and 194/SIP/FRP/E5/Dit.KI/VII/2017 and Pusat Penelitian Arkeologi Nasional export permit 2634/H5/TU/2017. For the skeletal material from Liang Bua, Liang Toge and Komodo, the research was undertaken as part of a collaborative project between Universitas Airlangga (Surabaya, Indonesia) and the Max Planck Institute for the Science of Human History (Jena, Germany) under the Kementerian Ristekdikti Research Permit 303/SIP/FRP/E5/Dit.KI/IX/2017 and Pusat Penelitian Arkeologi Nasional export permit 11404H5/TU/2017. Research of the human remains from Jareng Bori were carried out as part of a collaboration between Australian National University and Universitas Gaja Madja under the Kementerian Ristekdikti Research Permit 1209/FRP/E5/Dit.KI/VI/2016.

Note that full information on the approval of the study protocol must also be provided in the manuscript.

