## [Peer Review File · Nature Ecology & Evolution]

Peer Review Information

Journal: Nature Ecology & Evolution

Manuscript Title: Ancient genomes from the last three millennia support multiple human dispersals into Wallacea

Corresponding author name(s): Sandra Oliveira, Johannes Krause, Cosimo Posth, Mark Stoneking

Editorial Notes:

Reviewer Comments & Decisions:

Decision Letter, initial version:
--

14th December 2021

Dear Dr Oliveira,

Your manuscript entitled "Ancient genomes from the last three millennia support multiple human dispersals into Wallacea" has now been seen by three reviewers, whose comments are attached. The reviewers have raised a number of concerns which will need to be addressed before we can offer publication in Nature Ecology & Evolution. We will therefore need to see your responses to the criticisms raised and to some editorial concerns, along with a revised manuscript, before we can reach a final decision regarding publication.

We therefore invite you to revise your manuscript taking into account all reviewer and editor comments. Please highlight all changes in the manuscript text file [OPTIONAL: in Microsoft Word format].

* Include a "Response to reviewers" document detailing, point-by-point, how you addressed each

reviewer comment. If no action was taken to address a point, you must provide a compelling argument. This response will be sent back to the reviewers along with the revised manuscript.

* If you have not done so already please begin to revise your manuscript so that it conforms to our Article format instructions at <http://www.nature.com/natecolevol/info/final-submission>. Refer also to any guidelines provided in this letter.

[REDACTED]

Nature Ecology & Evolution is committed to improving transparency in authorship. As part of our efforts in this direction, we are now requesting that all authors identified as 'corresponding author' on published papers create and link their Open Researcher and Contributor Identifier (ORCID) with their account on the Manuscript Tracking System (MTS), prior to acceptance. ORCID helps the scientific community achieve unambiguous attribution of all scholarly contributions. You can create and link your ORCID from the home page of the MTS by clicking on 'Modify my Springer Nature account'. For more information please visit www.springernature.com/orcid.

[REDACTED]

Reviewer #3: Wallacean archaeology

Reviewers' comments:

Reviewer #1 (Remarks to the Author):

In the manuscript entitled "Ancient genomes from the last millennia support multiple human dispersals into Wallacea", Oliveira, Nägele, Carlhoff et al. provide a collection of 16 new ancient genomes from Wallacea (2600 to 250 years BP), including samples from Sulawesi, North Moluccas and East Nusa Tenggara. In this study, they focus on investigating variation in ancestry through time and space using principal component analysis, model-based clustering and f-statistics. Furthermore, the authors explored events of admixture and their associated timings that occurred in the region mainly between Papuan-related, East/Southeast Asian-related (associated with the Austronesian diaspora) and Mainland Southeast Asian-related sources. The main novelties of this manuscript include (i) the (early) genetic contribution from Mainland Southeast Asia to East Nusa Tenggara, which likely predated the arrival of Austronesian communities in the region, (ii) the complexity in the mode of admixture events (i.e. differences in the admixture history between islands, multiple or/and continuous in time), and (iii) the possible role of the North Moluccas in the subsequent dispersal to some parts of Remote Oceania.

The manuscript is well written, the population genetics and statistical analyses are thorough and there is no overinterpretation of the data. Indeed, the authors are extremely cautious in their conclusions, which are novel and highly interesting. This; together with the release of newly ancient genomes undoubtedly provide new insights into the history of the region. Importantly, this study reports ancient genomes from a region that is unsuitable for DNA preservation owing to its tropical nature, yet, these genomes are particularly important for understanding human population interactions and admixture in a region of extreme importance to understand the last massive human migrations to unsettled lands.

I find this study very good, 'to the point', well-thought, novel and of high interest for a general readership. Despite I have read it several times, I have no major comments but only a few suggestions and questions that, I believe, the authors should consider for the sake of clarity.

- Line 51. "[...] indicates occupation by AMH starting ..." ♦ I suggest to write the full name "anatomically modern humans" followed by the acronym.

- Lines 126-127. To visualize the evolution of genetic components through K values, I suggest to show in a supplementary figure, the results obtained for all values of K and indicate the cross-validation errors.

- Lines 141-143. "A significant positive result indicates" and "a significant negative result indicates": Even if the Z score threshold that you used to consider f4-statistics as significant is indicated in Figure S4, it would be helpful to mention it directly in the main text.

- Lines 144-145. "Our results show that ancient individuals from the North Moluccas share more drift with ancient individuals from Vanuatu (2900 BP) and Tonga (2500BP) than with Amis." Do the results remained significant when considering a more stringent Z score threshold (i.e. $|Z| > 3$)? One possibility would be to add a supplementary table for all f4-statistics with corresponding values, standard deviation and Z score

3Reviewer #2 (Remarks to the Author):

As the authors note in their introduction, understanding interactions between Austronesians and Indigenous pre-Austronesian populations will allow for a fuller understanding of the complex cultural interactions of the Australasian region. As such, this new information is very interesting to researchers working across Wallacea and surrounds. Do doubt this paper will provide some healthy data for discussions surrounding the changing cultures of Australasia.

Having said this, I think that the authors could probably bring out WHY this information is so informative better in their Discussion and even Abstract. At the moment, it feels like the team is too close to the data -- and has forgotten to clearly explain exactly why others should care about these findings. Its great to know how people interacted with each other in more detail -- but perhaps spell out a little more how this changes what we previously understood about people in this region? What does this change for archaeology? Adding a few lines on the differences between the northern vs. southern wallaceans may be a way forward here.

The paper provides a number of interesting insights into recent (over the past several thousand years) population movements and interactions, and as such, the authors don't need to "stoop" to bringing up that they retrieved aDNA from a particularly famous site (Liang Bua) when it has nothing to do with the H. florensiensis that that site is famous for. I felt that the sentences at the beginning of the Discussion (lines 227-228) "cheapened" their already interesting results.

Suggested Corrections:

There appears to be a mix of terms used to refer to Homo sapiens -- initially the term "Modern human" is used (line 48), but then the acronym (not introduced) of "AMH" is used on line 51, before "modern human" is returned to below -- Standardise throughout.

Introduction of acronym "MP" on line 324 that was not previously introduced in text.

"Evidence of" should be "Evidence for" -- grammar -- change throughout manuscript.

Reviewer #3 (Remarks to the Author):

The manuscript reports the first significant ancient DNA covering multiple regions of Indonesia and spanning a broad time period (~2600-250BP), very substantially increasing available aDNA data which has previously been based on samples from northern Sumatra and northeast Borneo, and the new Leang Panninge sample from Sulawesi, only. This is a very important contribution to aDNA coverage and research in the region. The authors argue for i) a composite Asian-related ancestry among ancient

4East Nusa Tenggara (NTT) samples that is not observed in the ancient North Moluccas (NM) samples , ii) that the mainland Southeast Asian ancestry signal in Nusa Tenggara pre-dates the arrival of genetic ancestry related to indigenous Taiwanese and Philippine groups and iii) that there are signals of ongoing contact and admixture. These inferences appear broadly consistent between analyses, but some clarifications are needed. While the ancient DNA is not show vastly different ancestry compared to modern regional groups, it does emphasize temporal dynamics, including the age of mainland SEA ancestry in NTT, and reveals more clearly aspects of regional ancestry variation. Overall, this paper is an important and interesting contribution to understanding demography in the region.

Major comments

Some separation of Asian ancestry in NM and NTT is supported by several analysis. Fig 1 is a successful introduction to ancestry patterns, but I think that the PCA project could be usefully extended, whether in SI or main text – the PC axes emphasize Asian/non-Asian ancestry and PNG/Solomon Islands ancestry, while the paper is largely about distinguishing closely related Asian ancestries. It may be the 3rd PC is more informative. Otherwise, a PCA projection including a single modern PNG population and the many Asian/Indonesian (but not Oceanian) samples would likely be useful in disentangling closely related ancestries.

f-statistic gradients. Given the small number of samples it is often unclear whether the linear fittings of the f-statistic biplots are significantly different. The analysis in Figure 2 would be stronger if it could be shown that e.g. a North Moluccas sample is outside the range of possible East Nusa Tenggara f_4 linear fits, taking into account uncertainty in individual sample f_4 statistics.

It isn't clear how confident aspects of the DATES analysis are. Including the admixture dates from a few relevant modern samples in Fig 4A (not lots of samples, e.g. earliest and latest dates from a region would be enough) would be helpful for this argument. Additionally, the uncertainty in the inference may not be sufficiently considered - in general, one standard error is shown rather than e.g. $1.96*SE$, and indeed one SE appears to be interpreted as a bound ('+-') in Fig S9. The difference in admixture dates inferred in NM between ancient and modern samples is nevertheless convincing. The difference between ancient and modern signals in NTT are more ambiguous (especially as modern dates appear to vary substantially) and this difference could be discussed and investigated further.

Given the proposed order of admixture in NTT, do DATES analysis of [Austronesian-proxy populations, mainland SEA populations] compared to [Austronesian-proxy populations, PNG populations] give similar results? Does a DATES analysis of [mainland SEA populations, PNG populations] give a slightly older date than [Austronesian-proxy populations, PNG populations], accepting that this is challenged by the Asian groups having qualitatively similar ancestry?

An analysis like TreeMix or qpGraph would be very helpful in teasing the differentiation between NM, Sulawesi and NTT apart.

More detail on the AHG analysis is needed rather than just the reference (to the Pugach et al paper?). E.g. in Table S6, negative values are selected in some cases but not others. While the qpAdm analysis

5does indeed support a 2-population mix in ancient samples and a 3-population mix in modern ones, clarification of the AHG analysis is needed.

Minor comments

Typo line 65 (', Papuan').

Indicate the aDNA sample code for sample level analyses (e.g. Table S4 and S5)

The abbreviation NTT (Fig 1B, S1) for East Nusa Tenggara presumably corresponds to Nusa Tenggara Timur. The abbreviation isn't explained in the text and is confusing given the use of the english East Nusa Tenggara throughout.

Mention that Fig 3 is qpAdm in the caption.

Suggest including the Sulawesi samples as well as the Bajo in Figure 1. The Bajo have unique lifestyle and depending on samples the ancestry may not be the best representation of Sulawesi generally.

"All pairs had the form f4(Mbuti, test; New Guinea Highlanders, ancient Wallacean) and always included Amis or ancient Vanuatu (2900 BP) vs. other Asian-related groups as test." – clarify wording?

*****END*****

Author Rebuttal to Initial comments

Dear Reviewers,

We are grateful for the attention that you devoted to our manuscript entitled "Ancient genomes from the last three millennia support multiple human dispersals into Wallacea". Below, we provide answers to your comments (in green) and outline the changes that have been taken to accommodate each suggestion.

Reviewer #1:

- Line 51. "[...] indicates occupation by AMH starting ..." I suggest to write the full name "anatomically modern humans" followed by the acronym.

6R: We now introduced the full name followed by the acronym in line 52, and use the acronym in line 56.

- Lines 126-127. To visualize the evolution of genetic components through K values, I suggest to show in a supplementary figure, the results obtained for all values of K and indicate the cross-validation errors.

R: Panels C-D of Figure S3 now include the DyStruct results for $K = 2$ to $K = 15$. To allow an easier inspection of labels and barplots, we still keep an expanded version of the results for the best K in panels E-F.

Unlike in ADMIXTURE, model evaluation in DyStruct is performed using the conditional log likelihood (CLL). However, similar to the cross-validation procedure, the CLL is obtained by holding out a subset of loci. The CLL results are shown in Figure S3A-B.

- Lines 141-143. "A significant positive result indicates" and "a significant negative result indicates": Even if the Z score threshold that you used to consider f_4 -statistics as significant is indicated in Figure S4, it would be helpful to mention it directly in the main text.

R: We added the Z score threshold to the main text (lines 150-151).

- Lines 144-145. "Our results show that ancient individuals from the North Moluccas share more drift with ancient individuals from Vanuatu (2900 BP) and Tonga (2500BP) than with Amis." Do the results remained significant when considering a more stringent Z score threshold (i.e $|Z| > 3$)? One possibility would be to add a supplementary table for all f_4 -statistics with corresponding values, standard deviation and Z score

R: Three out of the four f_4 -statistics that were significantly positive, remain positive with a Z score threshold of $Z > 3$. These results are now additionally reported in a new table (Table S3), as well as all other f_4 -statistic results presented in the manuscript (Table S4-6).

Reviewer #2:

As the authors note in their introduction, understanding interactions between Austronesians and Indigenous pre-Austronesian populations will allow for a fuller understanding of the complex cultural interactions of the Australasian region. As such, this new information is very interesting to researchers

7working across Wallacea and surrounds. Do doubt this paper will provide some healthy data for discussions surrounding the changing cultures of Australasia.

Having said this, I think that the authors could probably bring out WHY this information is so informative better in their Discussion and even Abstract. At the moment, it feels like the team is too close to the data -- and has forgotten to clearly explain exactly why others should care about these findings. Its great to know how people interacted with each other in more detail -- but perhaps spell out a little more how this changes what we previously understood about people in this region? What does this change for archaeology? Adding a few lines on the differences between the northern vs. southern wallaceans may be a way forward here.

R: To address this concern, we added a few changes to the abstract (lines 38-39) and some additional explanations to the discussion (lines 254-261 and 317-322).

The paper provides a number of interesting insights into recent (over the past several thousand years) population movements and interactions, and as such, the authors don't need to "stoop" to bringing up that they retrieved aDNA from a particularly famous site (Liang Bua) when it has nothing to do with the *H. floresiensis* that that site is famous for. I felt that the sentences at the beginning of the Discussion (lines 227-228) "cheapened" their already interesting results.

R: As our work has no direct relation to the *H. floresiensis* we removed the sentences that highlight the ancient DNA retrieved from the Liang Bua cave.

Suggested Corrections:

There appears to be a mix of terms used to refer to *Homo sapiens* -- initially the term "Modern human" is used (line 48), but then the acronym (not introduced) of "AMH" is used on line 51, before "modern human" is returned to below -- Standardise throughout.

R: These terms are now standardized (see lines 52 and 56).

Introduction of acronym "MP" on line 324 that was not previously introduced in text.

R: This acronym was defined in the introduction (line 69), but we now introduce it again in the discussion to add clarity (line 355).

"Evidence of" should be "Evidence for" -- grammar -- change throughout manuscript.

R: We corrected this in the manuscript.

Reviewer #3:

Major comments

Some separation of Asian ancestry in NM and NTT is supported by several analysis. Fig 1 is a successful introduction to ancestry patterns, but I think that the PCA project could be usefully extended, whether in SI or main text – the PC axes emphasize Asian/non-Asian ancestry and PNG/Solomon Islands ancestry, while the paper is largely about distinguishing closely related Asian ancestries. It may be the 3rd PC is more informative. Otherwise, a PCA projection including a single modern PNG population and the many Asian/Indonesian (but not Oceanian) samples would likely be useful in disentangling closely related ancestries.

R: We have extended our PCA analysis to provide a refined view of the relationship between the ancient Indonesian individuals and diverse Asian groups. Since PC3 and PC4 on the current datasets were still highly driven by Oceanian groups, we chose to follow the second suggestion and include only one population from PNG, while keeping the Indonesian and Asian-related populations. The results for PC1-3 are shown in a new Supplementary Figure (Figure S2). While the patterns emerging from the first two PC's highly resemble the previous PCA analysis (i.e. mostly driven by differences between Papuans and Asians), the plot of PC2 vs. PC3 provides a better view of the Asian ancestry relations. The modifications from lines 119 to 130 accommodate these new results.

f-statistic gradients. Given the small number of samples it is often unclear whether the linear fittings of the f-statistic biplots are significantly different. The analysis in Figure 2 would be stronger if it could be shown that e.g. a North Moluccas sample is outside the range of possible East Nusa Tenggara f4 linear fits, taking into account uncertainty in individual sample f4 statistics.

R: To evaluate differences in the linear fittings we applied a Bayesian approach that takes into account the measurement error in the f4-statistics (see methods, lines 504-526). We show that for several Southeast Asian test groups, the 95% credible interval of the differences does not overlap zero within the range of f4 values covering the Aru Manara 2150 BP, Tanjung Pinang 2100 BP, Uattamdi 1900 BP,

9Liang Bua 2600 BP, and Komodo 750 BP. This indicates that within that range there is a more than 95% chance that the observed F_4 -statistics for the East Nusa Tenggara and the North Moluccas differ. As we describe in the results (lines 169-182), the same cannot be confirmed below this range.

It isn't clear how confident aspects of the DATES analysis are. Including the admixture dates from a few relevant modern samples in Fig 4A (not lots of samples, e.g. earliest and latest dates from a region would be enough) would be helpful for this argument. Additionally, the uncertainty in the inference may not be sufficiently considered - in general, one standard error is shown rather than e.g. $1.96 \times SE$, and indeed one SE appears to be interpreted as a bound ('+') in Fig S9. The difference in admixture dates inferred in NM between ancient and modern samples is nevertheless convincing. The difference between ancient and modern signals in NTT are more ambiguous (especially as modern dates appear to vary substantially) and this difference could be discussed and investigated further.

R: We added an additional bar to Figure 4A representing the earliest and latest admixture dates for modern samples from each region (North Moluccas, Sulawesi, East Nusa Tenggara), as well as their respective lower and upper bound, now calculated as $2 \times SE$. We use $2 \times SE$ instead of $1.96 \times SE$ for consistency with the rest of the manuscript. Additionally, the bounds that we present in Figure S14 (previously Figure S9) now correspond to $2 \times SE$. The admixture time results for all modern populations are also provided in a new table (Table S11).

We did not claim that there is a difference between the admixture dates of present-day and ancient samples from East Nusa Tenggara. To make that clearer, we modified the main text (lines 236-242).

Given the proposed order of admixture in NTT, do DATES analysis of [Austronesian-proxy populations, mainland SEA populations] compared to [Austronesian-proxy populations, PNG populations] give similar results? Does a DATES analysis of [mainland SEA populations, PNG populations] give a slightly older date than [Austronesian-proxy populations, PNG populations], accepting that this is challenged by the Asian groups having qualitatively similar ancestry?

R: Estimating admixture using Austronesian vs mainland Southeast Asian populations as proxies is unfortunately not possible due to the low level of genetic differentiation between these two groups. If two populations chosen as sources of admixture do not show substantial differences in SNP allele frequencies, then for many SNPs in an admixed genome conclusive assignment to either of the admixing sources becomes problematic. In the original publication of ALDER (a similar method that is based on LD decay from multiple individuals, instead of ancestry covariance patterns from single individuals as used by DATES), the method was validated using parental sources with $F_{st}=0.15$, which for human SNP chip data corresponds to genetic distance between sub-Saharan Africa and Europe (Loh et al., 2013,

Genetics). The differentiation between the Austronesian and Southeast Asian groups is much more subtle ($F_{st} \sim 0.03-0.06$). As requested and to demonstrate the problem, we provide the covariance curves, obtained from admixture dating using Austronesian and mainland Southeast Asian populations as sources, which fail to detect admixture (we observe no exponential decay and the “curves” are flat).

To account for the second suggestion, we estimated admixture using as sources Austronesians vs. Papuans and mainland SEA vs. Papuans (new supplementary Figure S15). While the point estimates for the two Nusa Tenggara samples that have the most mainland SEA ancestry (Liang Bua 2600 BP and Komodo 750 BP) show older ages when mainland SEA is used as the Asian proxy, in agreement with our inferences on the order of admixture, the confidence intervals are largely overlapping. Therefore we cannot really say the admixture estimates are different. These results are reported in lines 243-249 and the details of the analysis were added to lines 551-556.

An analysis like TreeMix or qpGraph would be very helpful in teasing the differentiation between NM, Sulawesi and NTT apart.

R: We have conducted TreeMix and qpGraph analyses to model the relationships between different Wallaceans groups and groups from Asia and Oceania. However, in our explorations, we encountered two main challenges:

1) Due to the lack of ancient samples that could represent key streams of ancestry contributing to the genetic makeup of ancient Wallaceans (e.g. ancient Papuans), our models have to include both ancient individuals and modern populations. We found that many of the tested models recurrently show spurious associations between ancient samples or between ancient samples and an outgroup (e.g. graph in Figure R1; but note we found the same pattern in the TreeMix analysis), which might result from differences in residual amounts of ancient DNA damage. The removal of this association from a graph results in a rejection of the model ($|Z| > 3$). While in this particular example it might seem reasonable to assume that a contribution from Mbuti into the ancient Lapita-related individuals is very unlikely and therefore the association must be caused by some artifacts of the data, for more complex models the associations cannot be easily understood. These associations also put into question the validity of using Z-scores alone as a critical threshold for model acceptance/rejection and to guide the exploration of more complex scenarios, like those required to model ancient East Nusa Tenggara individuals.

Figure R1 - Testing the relationship between the ancient North Moluccas (Moluccas), ancient Lapita-related individuals (Vanuatu_Tonga), and present-day Kankanaey, Papuan Highlanders and Australians. Left: best-fitting graph. The worst-fitting Z score ($Z = 2.6$) corresponds to the f4-statistic $f_4(\text{Moluccas, Kankanaey, Moluccas, Australian})$. Right: removal of the admixture edge from Mbuti to Vanuatu_Tonga results in a rejection of the model, with the worst-fitting Z score ($Z = 4.1$) corresponding to the f4-statistic $f_4(\text{Mbuti, Papuan, Kankanaey, Vanuatu_Tonga})$. These graphs are based on >250,000 SNPs.

2) The modeling of the ancient East Nusa Tenggara individuals is also challenging because we currently lack a solid understanding of the relationships between all potential source groups. To overcome that, we first estimated the best-fitting graph based on different combinations of ancestry sources using the Markov chain Monte Carlo method implemented in AdmixtureBayes (Nielsen, 2018), which does not test a pre-defined graph but estimates the best graph directly from the data. Then, we compared the f-statistics calculated for the data with those predicted by the best-fitting graph using the qpGraph software. One drawback of the best-fitting graphs obtained through this procedure is that admixed groups are often placed in intermediate positions along the tree bifurcations that separate the two sources, instead of appearing as admixed. Such case is illustrated in Figure R2 by the Barito and ancient Nusa Tenggara. The convergence to this kind of topology when a group has allele frequencies intermediate between two other groups is not unexpected, due to the lower prior probability assigned to graphs with a higher number of admixture events (Nielsen, 2018). However, this model is at odds

with previous studies on western Indonesians, which showed that the Barito are an admixed group, having ancestry related to Southeast Asia and Austronesian groups. Therefore, we tested other potentially relevant graphs including additional admixture events and compared the Z-scores. Figure R3 shows the best graph that we obtained to represent the East Nusa Tenggara history. The worst-fitting Z-score ($Z = -4.35$) corresponds to the f4(ancient Nusa Tenggara, Australian; Tongan, ancient North Moluccas) which is above the commonly used threshold of $|Z| = 3$, implying that either Australians and Tongans or the ancient North Moluccas and Nusa Tenggara are more related to each other, than is suggested by the graph. In addition, this graph does not fully resolve the origins of the Southeast Asian-related ancestry in Nusa Tenggara, which comes out of a trifurcation rather than the branch that leads to the Southeast Asian source (Nicobarese). It is likely that this pattern emerges from an insufficient model for the Southeast Asian source. For example, Lipson et al., 2018 (Science, Supplementary figures) presents a very complex model of the history of Nicobarese, but also for Man Bac and Mlabri (groups also tested by us). While we could certainly increase the complexity of our models until we find a better fit for the Nusa Tenggara individuals, we do not think such models will provide a clearer view on the Nusa Tenggara history. Additionally, for very complex models the number of possible configurations of the graph increases substantially and there is no robust test yet designed to compare the likelihood of such graphs.

To sum up, we decided not to include the results of the qpGraph and Treemix analyses in the manuscript due to our concerns about their helpfulness in modeling regions known to have a complex history and about potential biases in combining modern and ancient samples in the same analysis. We also prefer to avoid adding too many technical considerations on qpGraph methods, which would be required to explain the observed patterns, and would lead to an extension of the manuscript text with no obvious benefit. Moreover, many other ancient DNA studies do not include such analyses (e.g. Schuenemann et al., 2017, Nature communications; Amorim et al., 2018, Nature communications; Bongers et al., 2020, PNAS; Fernandes et al., 2020, Nature Ecology and Evolution), and we think the various analyses included in the manuscript provide strong support for the additional Southeast Asian ancestry in the East Nusa Tenggara.

Figure R2 - Exploring the admixture history of ancient North Moluccas and ancient Nusa Tenggara individuals. The best graph identified by AdmixtureBayes for the subset of groups presented here places the Asian ancestry of ancient individuals from Nusa Tenggara, as well as Barito (Western Indonesia), in an intermediate position between the split of Mlabri (Southeast Asian) and the split of Austronesian-

related groups (Kankanaey). The graph was build with ~400,000 SNPs and has a worst-fitting Z-score of $Z = -8.83$.

Figure R3 - The best-fitting admixture graph found as described in the text for ancient individuals from the North Moluccas and Nusa Tenggara. The graph was build with ~400,000 SNPs and has a worst-fitting Z-score of $Z = -4.35$.

More detail on the AHG analysis is needed rather than just the reference (to the Pugach et al paper?). E.g. in Table S6, negative values are selected in some cases but not others. While the qpAdm analysis does indeed support a 2-population mix in ancient samples and a 3-population mix in modern ones, clarification of the AHG analysis is needed.

R: We added a short summary of the AHG analysis to the methods section (lines 540-549) and also indicate that the sequence of admixture events was determined by the configuration that produced the smallest absolute value of the covariance (i.e. the value closer to zero).

Minor comments

Typo line 65 (' , Papuan').

R: We removed the comma.

Indicate the aDNA sample code for sample level analyses (e.g. Table S4 and S5)

R: This has been added (Table S8 and S9 of the current version).

The abbreviation NTT (Fig 1B, S1) for East Nusa Tenggara presumably corresponds to Nusa Tenggara Timur. The abbreviation isn't explained in the text and is confusing given the use of the english East Nusa Tenggara throughout.

R: We added a note to the Figure legends.

Mention that Fig 3 is qpAdm in the caption.

R: This has been added.

Suggest including the Sulawesi samples as well as the Bajo in Figure 1. The Bajo have unique lifestyle and depending on samples the ancestry may not be the best representation of Sulawesi generally.

R: We could not include the Sulawesi sample in Figure 1 because that would require merging data from three different SNP arrays, resulting in an insufficient number of SNPs for PCA. The dataset used to compute the PCA of Figure 1 consists of publicly available whole genome data merged with Human Origins and Affymetrix 6.0 genotype data, and only includes one population from Sulawesi – the Bajo. The other Sulawesi sample, as well as Bajo, can be found in Figure S1, which used another dataset merging publicly available whole genome data merged with Human Origins and Affymetrix Axiom Genome-Wide Human genotype data.

“All pairs had the form $f_4(\text{Mbuti, test; New Guinea Highlanders, ancient Wallacean})$ and always included Amis or ancient Vanuatu (2900 BP) vs. other Asian-related groups as test.” – clarify wording?

R: This sentence was modified (lines 163-166).

Additional modifications:

In addition to the reviewer’s suggestions, we have added some modifications to accommodate two new references (discussion: lines 288-290 and 335-337) and to improve clarity. All modifications are highlighted in yellow in the main text.

Decision Letter, first revision:

4th March 2022

Dear Dr. Oliveira,

Thank you for submitting your revised manuscript "Ancient genomes from the last three millennia support multiple human dispersals into Wallacea" (NATECOLEVOL-211014965A). It has now been seen again by the original reviewers and their comments are below. The reviewers find that the paper has improved in revision, and therefore we'll be happy in principle to publish it in Nature Ecology &

15Evolution, pending minor revisions to satisfy the reviewers' final requests and to comply with our editorial and formatting guidelines.

[REDACTED]

Reviewer #1 (Remarks to the Author):

This is a revised version of a manuscript I previously reviewed. The authors have satisfactorily addressed all my previous concerns or answer to my questions. For me, this manuscript is ready to be published.

Reviewer #2 (Remarks to the Author):

The authors have sufficiently addressed each of my previous comments in their revised version.

Reviewer #3 (Remarks to the Author):

The authors have been diligent in refining analyses as suggested. These additions are helpful, it is good to see uncertainty taken into account in several analyses, and some additional comments on interpretation.

Several proposed extensions had technical issues that likely highlight limitations of certain methods. I agree with the authors that the general results of the paper likely stand nonetheless, and do not think that including e.g. qpGraphs is required. The best fitting graphs deviate significantly from the data but importantly do appear to support overall differences between the two Wallacean groups.

The minor points have also been corrected.

Our ref: NATECOLEVOL-211014965A

8th March 2022

Dear Dr. Oliveira,

Thank you for your patience as we've prepared the guidelines for final submission of your Nature Ecology & Evolution manuscript, "Ancient genomes from the last three millennia support multiple human dispersals into Wallacea" (NATECOLEVOL-211014965A). Please carefully follow the step-by-step instructions provided in the attached file, and add a response in each row of the table to indicate the changes that you have made. Please also check and comment on any additional marked-up edits we have proposed within the text. Ensuring that each point is addressed will help to ensure that your revised manuscript can be swiftly handed over to our production team.

****We would like to start working on your revised paper, with all of the requested files and forms, as soon as possible (preferably within two weeks). Please get in contact with us immediately if you anticipate it taking more than two weeks to submit these revised files.****

In recognition of the time and expertise our reviewers provide to Nature Ecology & Evolution's editorial process, we would like to formally acknowledge their contribution to the external peer review of your manuscript entitled "Ancient genomes from the last three millennia support multiple human dispersals into Wallacea". For those reviewers who give their assent, we will be publishing their names alongside the published article.

Nature Ecology & Evolution offers a Transparent Peer Review option for new original research manuscripts submitted after December 1st, 2019. As part of this initiative, we encourage our authors to support increased transparency into the peer review process by agreeing to have the reviewer comments, author rebuttal letters, and editorial decision letters published as a Supplementary item. When you submit your final files please clearly state in your cover letter whether or not you would like to participate in this initiative. Please note that failure to state your preference will result in delays in accepting your manuscript for publication.

17Cover suggestions

As you prepare your final files we encourage you to consider whether you have any images or illustrations that may be appropriate for use on the cover of Nature Ecology & Evolution.

Nature Ecology & Evolution has now transitioned to a unified Rights Collection system which will allow our Author Services team to quickly and easily collect the rights and permissions required to publish your work. Approximately 10 days after your paper is formally accepted, you will receive an email in providing you with a link to complete the grant of rights. If your paper is eligible for Open Access, our Author Services team will also be in touch regarding any additional information that may be required to arrange payment for your article.

Please note that *Nature Ecology & Evolution* is a Transformative Journal (TJ). Authors may publish their research with us through the traditional subscription access route or make their paper immediately open access through payment of an article-processing charge (APC). Authors will not be required to make a final decision about access to their article until it has been accepted. [Find out more about Transformative Journals](https://www.springernature.com/gp/open-research/transformative-journals)

Authors may need to take specific actions to achieve [compliance with funder and institutional open access mandates](https://www.springernature.com/gp/open-research/funding/policy-compliance-faqs). If your research is supported by a funder that requires immediate open access (e.g. according to [Plan S principles](https://www.springernature.com/gp/open-research/plan-s-compliance)) then you should select the gold OA route, and we will direct you to the compliant route where possible. For authors selecting the subscription publication route, the journal's standard licensing terms will need to be accepted, including [self-archiving-and-license-to-publish](https://www.nature.com/nature-portfolio/editorial-policies/self-archiving-and-license-to-publish). Those licensing terms will supersede any other terms that the author or any third party may assert apply to any version of the manuscript.

For information regarding our different publishing models please see our <https://www.springernature.com/gp/open-research/transformative-journals> Transformative Journals page. If you have any questions about costs, Open Access requirements, or our legal forms, please contact ASJournals@springernature.com.

[REDACTED]

[REDACTED]

Reviewer #1:

Remarks to the Author:

This is a revised version of a manuscript I previously reviewed. The authors have satisfactorily addressed all my previous concerns or answer to my questions. For me, this manuscript is ready to be published.

Reviewer #2:

Remarks to the Author:

The authors have sufficiently addressed each of my previous comments in their revised version.

Reviewer #3:

Remarks to the Author:

The authors have been diligent in refining analyses as suggested. These additions are helpful, it is good to see uncertainty taken into account in several analyses, and some additional comments on interpretation.

Several proposed extensions had technical issues that likely highlight limitations of certain methods. I agree with the authors that the general results of the paper likely stand nonetheless, and do not think that including e.g. qpGraphs is required. The best fitting graphs deviate significantly from the data but importantly do appear to support overall differences between the two Wallacean groups.

The minor points have also been corrected.

19Final Decision Letter:

13th April 2022

Dear Dr Oliveira,

We are pleased to inform you that your Article entitled "Ancient genomes from the last three millennia support multiple human dispersals into Wallacea", has now been accepted for publication in Nature Ecology & Evolution.

Over the next few weeks, your paper will be copyedited to ensure that it conforms to Nature Ecology and Evolution style. Once your paper is typeset, you will receive an email with a link to choose the appropriate publishing options for your paper and our Author Services team will be in touch regarding any additional information that may be required

You will not receive your proofs until the publishing agreement has been received through our system

Due to the importance of these deadlines, we ask you please us know now whether you will be difficult to contact over the next month. If this is the case, we ask you provide us with the contact information (email, phone and fax) of someone who will be able to check the proofs on your behalf, and who will be available to address any last-minute problems . Once your paper has been scheduled for online publication, the Nature press office will be in touch to confirm the details.

Acceptance of your manuscript is conditional on all authors' agreement with our publication policies (see www.nature.com/authors/policies/index.html). In particular your manuscript must not be published elsewhere and there must be no announcement of the work to any media outlet until the publication date (the day on which it is uploaded onto our web site).

Please note that *Nature Ecology & Evolution* is a Transformative Journal (TJ). Authors may publish their research with us through the traditional subscription access route or make their paper immediately open access through payment of an article-processing charge (APC). Authors will not be required to make a final decision about access to their article until it has been accepted. [Find out more about Transformative Journals](https://www.springernature.com/gp/open-research/transformative-journals)

Authors may need to take specific actions to achieve [compliance](https://www.springernature.com/gp/open-research/funding/policy-compliance-faqs) with funder and institutional open access mandates. If your research

20is supported by a funder that requires immediate open access (e.g. according to [Plan S principles](https://www.springernature.com/gp/open-research/plan-s-compliance)) then you should select the gold OA route, and we will direct you to the compliant route where possible. For authors selecting the subscription publication route, the journal's standard licensing terms will need to be accepted, including <https://www.nature.com/nature-portfolio/editorial-policies/self-archiving-and-license-to-publish>. Those licensing terms will supersede any other terms that the author or any third party may assert apply to any version of the manuscript.

We welcome the submission of potential cover material (including a short caption of around 40 words) related to your manuscript; suggestions should be sent to Nature Ecology & Evolution as electronic files (the image should be 300 dpi at 210 x 297 mm in either TIFF or JPEG format). Please note that such pictures should be selected more for their aesthetic appeal than for their scientific content, and that colour images work better than black and white or grayscale images. Please do not try to design a cover with the Nature Ecology & Evolution logo etc., and please do not submit composites of images related to your work. I am sure you will understand that we cannot make any promise as to whether any of your suggestions might be selected for the cover of the journal.

You can generate the link yourself when you receive your article DOI by entering it here: <http://authors.springernature.com/share>.

[REDACTED]

21P.S. Click on the following link if you would like to recommend Nature Ecology & Evolution to your librarian <http://www.nature.com/subscriptions/recommend.html#forms>

** Visit the Springer Nature Editorial and Publishing website at http://editorial-jobs.springernature.com?utm_source=ejp_NEcoE_email&utm_medium=ejp_NEcoE_email&utm_campaign=ejp_NEcoE for more information about our career opportunities. If you have any questions please click [here](mailto:editorial.publishing.jobs@springernature.com).**